# POST-TRAINING WEIGHTED QUANTIZATION OF NEURAL NETWORKS FOR LANGUAGE MODELS

## ABSTRACT

As a practical model compression technique, parameter quantization is effective especially for language models associated with a large memory footprint. Neural network quantization is usually performed to reduce quantization loss assuming that quantization error of each parameter equally contributes to the overall training loss. The importance of each parameter, however, may highly differ such that for the same number of quantization bits, certain parameters lead to higher training loss than the others after quantization. In this paper, we consider a non-uniform quantization scheme, specifically binary-coding-based quantization, for high compression ratio and efficient computations while avoiding large accuracy degradation by uniform quantization (e.g., INT8). Then, we derive quantization optimization methods to take into account the importance of each parameter. We demonstrate that for post-training quantization, weight magnitude can represent importance and improve model accuracy significantly compared to the previous schemes lacking importance considerations. For various language models including BERT, DistilBERT, AWD-LSTM, and Transformer, we achieve 2-4 bits per weight by our proposed post-training quantization with reasonable accuracy degradation.

## 1 INTRODUCTION

Training techniques for deep neural networks (DNNs) have been developed in ways to incur a lot of parameter redundancy to expedite seeking local minima (Denil et al., 2013; Jonathan Frankle, 2019). As a result, various model compression techniques including parameter pruning (Han et al., 2015; He et al., 2017), quantization (Courbariaux et al., 2015; Rastegari et al., 2016), low-rank approximation (N. Sainath et al., 2013; Prabhavalkar et al., 2016), and knowledge distillation (Hinton et al., 2015; Polino et al., 2018) are proposed to lower storage requirements and improve inference performance. Several compression techniques can be combined in a synergistic way to enhance compression ratio (Han et al., 2016; Zhu et al., 2017). In this work, we consider parameter quantization that maintains structured model formats and presents high compression ratio. Note that due to limited hardware resources, quantization is an essential method for any inference systems. In general, quantization is classified into uniform quantization based on fixed-point parameter representations (Jacob et al., 2018; Han et al., 2016) and non-uniform quantization associated with the binary codes (Zhou et al., 2017; Rastegari et al., 2016) or codebooks (Choi et al., 2017; Stock et al., 2020).

Most DNN quantization methods are performed based on the principle of minimizing the mean squared error (MSE) of quantized parameters (Rastegari et al., 2016; Xu et al., 2018; Zhou et al., 2017). Optimizing the MSE is also an underlying principle of low-rank approximation techniques such as the singular value decomposition (SVD) (Prabhavalkar et al., 2016; N. Sainath et al., 2013). Note that, however, minimizing the MSE implies that each parameter is equally important (i.e., squared errors from parameters are accumulated without considering importance of each weight). In practice, the impact of each parameter perturbation from quantization on training loss can be vastly different and such impact needs to be analyzed through a sensitivity study of each parameter toward a change in training loss value. In other words, minimizing the MSE (or the Euclidean distance between original parameters and quantized parameters) may not correspond to minimizing training loss function after quantization.

Robustness to quantization error of each parameter can be expressed as sensitivity. Sensitivity of $i$-th parameter $w_i$ is the amount of change in the loss function when $w_i$ is perturbed. A parameter

associated with high sensitivity would require relatively smaller quantization error if quantization is performed in a group manner. Several previous works acknowledge distinct sensitivity of each parameter to improve quantization quality. Note that because exact sensitivity estimation of each parameter toward loss function is highly complicated, various heuristic techniques have been introduced. For example, Hessian-weighted k-means clustering is used for codebook-based implementations (Choi et al., 2017) or Taylor series expansion to bound loss function difference is conducted to decide the optimal quantization bits of each weight (Khoram & Li, 2018). The Hessian matrix can be used to assign different numbers of quantization bits for each layer (Dong et al., 2019; Shen et al., 2019). Minimizing the reconstruction error on the output activations after each layer quantization is performed in (Stock et al., 2020).

In this paper, we propose a weighted quantization framework where quantized parameters follow the structure of the binary codes so as to achieve high compression ratio and high computational efficiency (Rastegari et al., 2016; Jeon et al., 2020). Specifically, given that an importance of each parameter is represented as a real number between 0 and 1, we extract an optimal quantization solution modified from the previous binary-coding-based quantization methods that employ equal parameter importance. Similar to previous attempts, we also find that calculating accurate importance of each parameter is challenging. As a successful approximation of importance, we suggest that magnitude-based importance estimation is especially effective for post-training non-uniform quantization.

## 2 POST-TRAINING PARAMETER QUANTIZATION FOR LANGUAGE MODELS

The number of parameters for language models is dramatically increasing (e.g., GPT-3 (Brown et al., 2020) requires 175 billion parameters). Correspondingly, model compression for language models is becoming a mandatory process to reduce response time and inference energy. We devise a compression method considering the followings:

- Recent language models are usually memory-bound because of small batch size and lacking layers of high reuse (e.g., conv layers). Thus, **reducing memory footprint is critical**.

- Compression algorithms should be supported by **dedicated kernels**, designed specifically for language models if possible.

- Compression-aware training is challenging and expensive if hyper-parameters are added to already huge language models (hence, **we choose a post-training method.**)

Fixed-point inference using uniform quantization is not desirable for language models because of noticeable accuracy degradation (Shen et al., 2019; Jeon et al., 2020) while the advantage of small computational units (e.g., INT8 MAC) is insignificant for memory-bound applications. Thus, we adopt float-based parameter quantization (i.e., expected values of quantized parameters remains to be of full precision) that induce a lot smaller number of quantization bits compared to fixed-point quantization (Xu et al., 2018; Stock et al., 2020).

Recently, a kernel library, called `BiQGEMM` (Jeon et al., 2020), was introduced to support binary-coding-based quantization techniques to accelerate quantized neural networks. Using lookup tables, `BiQGEMM` enables byte-level memory accesses and achieves $8.3\times$ run-time memory footprints and $3.5\times$ speed up with a mobile CPU for Transformer (Chung et al., 2020). As a result, binary-coding-based quantization has become a practical approach to quantizing language models. As such, *we restrict our interests to **binary-coding-based quantization technique** in this paper*.

Quantization-aware training is an active research area to improve model accuracy (Courbariaux et al., 2015; Lee et al., 2018). We note that in the case of language models, however, there are numerous occasions when retraining for quantization is not available. For example, quantization-aware training requires in-depth knowledge on model compression while model designers may not have such expertise. On the other hand, the original training code or the entire training data may not be shared with model compression engineers. Also, modifying the original DNN models to be aware of quantization would increase model design efforts and training time significantly. Since language models already demand significant training time and cost, adding additional training complexity by quantization-aware training would not be a practical option. As such, post-training quantization without retraining is gaining increasing attention (Zhao et al., 2019; Nagel et al., 2019).

# 3 WEIGHTED QUANTIZATION BASED ON THE BINARY CODES

As discussed, we choose post-training binary-coding-based quantization as our strategy to compress language DNN models efficiently. Following the Binary-Weight-Networks (Rastegari et al., 2016) introducing the binary codes as a quantization format, a weight vector $\boldsymbol{w}$ is approximated to be $\alpha \boldsymbol{b}$ by using a scaling factor $\alpha \in \mathbb{R}$ and a binary vector $\boldsymbol{b}\, (\in \{-1, +1\}^n)$, where $n$ is the vector size. A real number scaling factor is shared by multiple weights such that binary vector $\boldsymbol{b}$ occupies most of the weight storage requirements. Binary codes eliminate the need for dequantization for inference, leading to reduced on-chip memory size for weights. In this section, we study general weighted quantization methodologies when quantization follows the format of the binary codes and and quantization error recognizes sensitivity information.

## 3.1 GREEDY METHOD AND ALTERNATING METHOD WITHOUT IMPORTANCE CONSIDERATIONS

In general, non-uniform weight quantization methods (in the form of the binary codes) strive to minimize $\|\boldsymbol{w} - \alpha \boldsymbol{b}\|^2$. In the case of 1-bit quantization, we obtain the following analytical solution:

$$\boldsymbol{b}^* = \text{sign}(\boldsymbol{w}),\ \alpha^* = \frac{\boldsymbol{w}^\top \boldsymbol{b}^*}{n}. \tag{1}$$

On the other hand, in the case of multi-bit quantization, there is no analytical solution (Rastegari et al., 2016; Xu et al., 2018). As a result, various approximated methods exist for multi-bit quantization.

**Greedy Method**   As a computationally simple method, 1-bit quantization shown in Eq. (1) can be extended to multi-bit ($q$-bit) quantization (Guo et al., 2017). Specifically, $i^\text{th}$-bit ($i > 1$) quantization is performed by minimizing the residue of $(i-1)^\text{th}$-bit quantization as following:

$$\min_{\alpha_i, \boldsymbol{b}_i} \|\boldsymbol{r}_{i-1} - \alpha_i \boldsymbol{b}_i\|^2,\ \text{where}\ \boldsymbol{r}_{i-1} = \boldsymbol{w} - \sum_{j=1}^{i-1} \alpha_j \boldsymbol{b}_j,\ 1 < i \le q. \tag{2}$$

The optimal solution of Eq. (2) is then given as

$$\boldsymbol{b}_i^* = \text{sign}(\boldsymbol{r}_{i-1}),\ \alpha_i^* = \frac{\boldsymbol{r}_{i-1}^\top \boldsymbol{b}_i^*}{n}. \tag{3}$$

**Alternating Method**   Greedy method described above is non-iterative. In order to reduce $\|\boldsymbol{w} - \sum_{i=1}^q \alpha_i \boldsymbol{b}_i\|^2$ further than Greedy method, iterative methods would be necessary while increasing the number of iterations tends to lower quantization error. Once initial $\alpha$ and $\boldsymbol{b}$ values are calculated by Greedy method, one can notice that $\{\alpha_i\}_{i=1}^q$ can be refined (Guo et al., 2017) as

$$[\alpha_1, ..., \alpha_q] = \left( \left( \boldsymbol{B}_q^\top \boldsymbol{B}_q \right)^{-1} \boldsymbol{B}_q^\top \boldsymbol{w} \right)^\top,\ \text{when}\ \boldsymbol{B}_q = [\boldsymbol{b}_1, ..., \boldsymbol{b}_q] \in \{-1, +1\}^{n \times q}. \tag{4}$$

Then, $\boldsymbol{B}_q$ can be refined as well by binary search given a new refined $\{\alpha_i\}_{i=1}^q$. As a result, $\{\alpha_i\}_{i=1}^q$ and $\boldsymbol{B}_q$ are refined alternatively. Alternating refinements of $\{\alpha_i\}_{i=1}^q$ and $\boldsymbol{B}_k$ are repeated until there is no noticeable improvement in quantization error. Such iterative quantization procedure is introduced as Alternating multi-bit method (Xu et al., 2018).

## 3.2 IMPORTANCE-AWARE WEIGHTED QUANTIZATION

Let us assume that the importance of the $i$-th parameter is normalized and given as $m_i$ ($0 \le m_i \le 1$). Then, we minimize the weighted quantization loss $\sum_{i=1}^n (m_i(w_i - \hat{w}_i)^2)$ where $w_i$ is quantized to be $\hat{w}_i = \sum_{j=1}^q (\alpha_j b_j)$. Before studying how to estimate importance values, we are interested in finding modified versions of Greedy method and Alternating method when importance values are given. For 1-bit quantization, weighted quantization also has the following analytical solution:

$$\boldsymbol{b}^* = \text{sign}(\boldsymbol{w}),\ \alpha^* = \frac{\sum_{i=1}^n (m_i |w_i|)}{\sum_{i=1}^n m_i}. \tag{5}$$

Note that if all importance values are equal (e.g., $m_i = 1$ for all $i$), then Eq. (5) becomes the same as Eq. (1). Correspondingly, Eq. (1) can be regarded as a special case of Eq. (5). Compared to the conventional Greedy method, our proposed importance-aware Greedy method demands modifications to $\alpha_i$ calculations as $\alpha_i^* = \sum_{i=1}^{n}(m_i|r_{i-1}|)/\sum_{i=1}^{n} m_i$.

For the importance-aware Alternating method, we first conduct the importance-aware Greedy method. Then, Eq. (4) is to be transformed to employ importance. Let us define an $n$-by-$n$ diagonal matrix $\boldsymbol{M} = \text{diag}(m_1, ..., m_n)$, where each diagonal element is an importance value $m_i$. By solving linear least squares, $\alpha$ values are refined as

$$[\alpha_1, ..., \alpha_q] = \left( \left( \boldsymbol{B}_q^\top \boldsymbol{M} \boldsymbol{B}_q \right)^{-1} \boldsymbol{B}_q^\top \boldsymbol{M} \boldsymbol{w} \right)^\top, \text{ when } \boldsymbol{B}_q = [\boldsymbol{b}_1, ..., \boldsymbol{b}_q] \in \{-1, +1\}^{n \times q}, \quad (6)$$

while refining $\boldsymbol{B}_q$ is still performed by binary search using refined scaling factors. Accordingly, Eq. (4) is a particular case of Eq. (6) when $\boldsymbol{M}$ is an identity matrix.

Overall, our proposed importance-aware quantization scheme is comprehensive to include previous methods as a subset. In the rest of this paper, we investigate simple and efficient schemes to estimate importance metrics applicable to post-training non-uniform binary-coding-based quantization.

## 4 IMPORTANCE ESTIMATION USING WEIGHT MAGNITUDE

Sensitivity or importance of each parameter can be estimated by evaluating the loss function change induced by parameter perturbation. Such estimation, however, is computationally demanding since each parameter's perturbation requires computations of entire feedforward paths while the number of parameters is ever increasing in recent DNN designs. Moreover, it is difficult to decide an appropriate amount of perturbation of parameters. In this section, we propose an efficient importance estimation method based on weight magnitude for fast and high-accurate post-training weight quantization.

### 4.1 WEIGHT MAGNITUDE AS IMPORTANCE

As an effort to study a factor affecting importance, we introduce Optimal Brain Damage (OBD) (LeCun et al., 1990). OBD approximates the loss function by the Taylor series and a perturbation of the loss function $\delta L$ by weight perturbations is presented as

$$\delta L = \sum_{i=1}^{v} \frac{\partial L}{\partial w_i} \delta w_i + \frac{1}{2} \sum_{i=1}^{v} h_{i,i} \delta w_i^2 + \frac{1}{2} \sum_{i=1}^{v} \sum_{\substack{j=1 \\ j \neq i}}^{v} h_{i,j} \delta w_i \delta w_j + O(\|\delta w\|^3), \quad (7)$$

where $v$ is the number of weights and $h_{i,j}$ is an element of the Hessian matrix. Note that at a local minimum, the first term is eliminated while all the $h_{i,i}$ are non-negative. Using "diagonal" approximation and "quadratic" approximation (LeCun et al., 1990), Eq. (7) is simplified as

$$\delta L \simeq \frac{1}{2} \sum_{i=1}^{v} h_{i,i} \delta w_i^2. \quad (8)$$

As described in Eq. 8, diagonal elements of the Hessian matrix can be used as effective importance metrics for loss-aware training algorithms for quantization (Hou et al., 2017; Hou & Kwok, 2018). Note that for post-training quantization, unfortunately, the first or the second partial derivatives (i.e., the gradient or the Hessian) are not available. Thus, for post-training quantization, importance is required to be given as a function of weight magnitude. In other words, we are interested whether Hessian-based importance can be replaced with magnitude-based importance. **The goal of this work is then to empirically show that magnitude-based importance is indeed practical for post-training binary-coding-based quantization.**

To verify our basic assumption that large weights would present higher importance for binary-coding-based quantization, we controlled scaling factors (i.e., $\alpha$ values of the minimum MSE are multiplied by 'Scaling Factor Multiplier') of a layer in an LSTM model using PTB dataset (see details of models and dataset in Appendix) as shown in Figure 1. Indeed, when scaling factors become larger than the ones obtained by minimizing MSE present, test perplexity is enhanced for both layers.

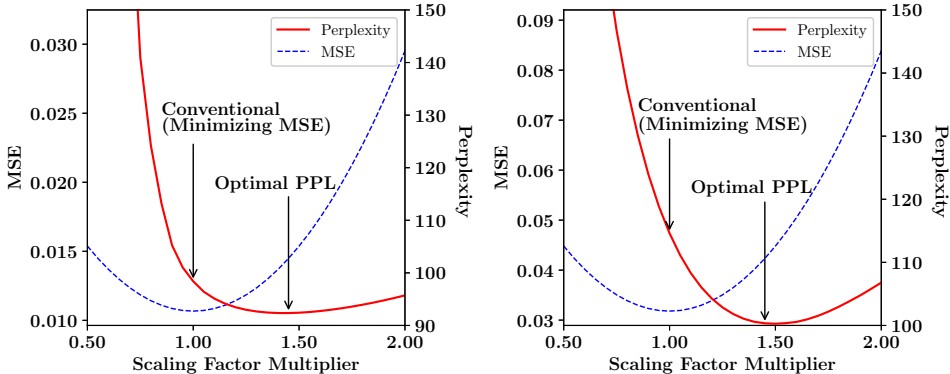

Figure 1: Quantization error (MSE) and test perplexity when one selected layer of an LSTM model using PTB dataset is quantized to be 1-bit. (Left): Embedding layer. (Right): LSTM layer.

## 4.2 HYPER-PARAMETERS FOR IMPORTANCE METRICS

The underlying principle of our importance estimation methodology is that relatively smaller quantization errors are supposed to be applied to weights of larger magnitude. To fine-tune our proposed importance estimation scheme, we propose the following three hyper-parameters: 1) an exponent to control a correlation between importance and magnitude of a weight, 2) a parameter to clip importance in order to handle outliers in magnitude distributions, and 3) a pruning parameter to exclude weights of low magnitude during quantization optimization.

$E$ (**Exponent**)    A basic form to calculate normalized importance of each weight is presented as

$$m_i = \left| \frac{w_i}{w_{max}} \right|^E, \tag{9}$$

where $w_{max}$ is the maximum weight magnitude in a given layer (i.e., $m_i$ is computed in a layer-wise manner) and the weight of the largest magnitude is considered to be the most important. A value between 0.0 and 1.0 for $E$ is primarily adopted in our experiments to yield sub-linear importance increase as weight magnitude increases. $E = 0$ results in the conventional quantization method with $m_i = 1$ for all $w_i$.

$C$ (**Clipping Importance**)    For a given distribution of weight magnitude, a few large outliers may distort the entire distribution of $m_i$ obtained by Eq. (9). In other words, because of a few exceptionally large weights, most weights may exhibit small importance values. A conventional quantization technique to prevent outliers in a distribution is to clip weights/activations and/or gradients (Choi et al., 2018; Zhao et al., 2019; Goodfellow et al., 2016). For Eq. (9), $w_{max}$ is decided to be the weight magnitude at the $(C \times 100)$-th percentile when $0 < C \leq 1$. If $m_i$ exceeds 1.0, then $m_i$ is forced to be 1.0 (note that $w_i$ is not clipped).

$P$ (**Pruning for Quantization**)    Due to regularization effects, a lot of weights employ small magnitudes and a weight distribution usually follows a Gaussian distribution (Goodfellow et al., 2016). As a result, a large number of small weights (with less importance as in Eq. (8)) may take a large portion of total quantization error unless $m_i$ values of those weights are extremely small. Note that even though pruning prior to quantization is an effective method to improve quantization quality (Li & Liu, 2016; Zhu et al., 2017), pruning would yield one additional bit per weight for masking information or sparse matrix formats with low parallelism for DNN inference. In our work, 1) we exclude weights of magnitude smaller than the $(P \times 100)$-th percentile from the quantization optimization, 2) find the scaling factors and the binary codes using weights larger than the $(P \times 100)$-th percentile, and 3) all of the excluded small weights are assigned to a binary code with the smallest magnitude available from combining scaling factors (while each sign information is maintained). Accordingly, while we adopt a parameter pruning idea for quantization, additional pruning mask data is not necessary. In short, small weights are not considered while obtaining the binary codes, and then, replaced with the smallest weight in the binary codes.

Table 1: Post-training (3 bits per weight) quantization comparison on MSE (quantization error), average scaling factor values, training loss, and training model accuracy. For importance metrics, $E = 1.0$ is used while $P$ and $C$ are not considered.

| Model | Method | Quant. Err. (MSE) | Average Scaling Factors | Training Loss | Training Accuracy (%) |
|---|---|---|---|---|---|
| BERT (MRPC) | Greedy (No Importance) | 1.13e-04 | 0.021 | 0.405 | 83.70 |
| | Greedy ($E$=1.0) | 1.01e-04 | 0.028 | 0.370 | 84.71 |
| | Alter. (No Importance) | 6.79e-05 | 0.029 | 0.693 | 80.81 |
| | Alter. ($E$=1.0) | 1.05e-04 | 0.035 | 0.212 | 93.13 |
| BERT (MNLI) | Greedy (No Importance) | 1.14E-04 | 0.021 | 0.726 | 76.32 |
| | Greedy ($E$=1.0) | 1.02E-04 | 0.029 | 0.418 | 86.93 |
| | Alter. (No Importance) | 6.84E-05 | 0.028 | 0.349 | 89.75 |
| | Alter. ($E$=1.0) | 1.07E-04 | 0.035 | 0.282 | 91.85 |

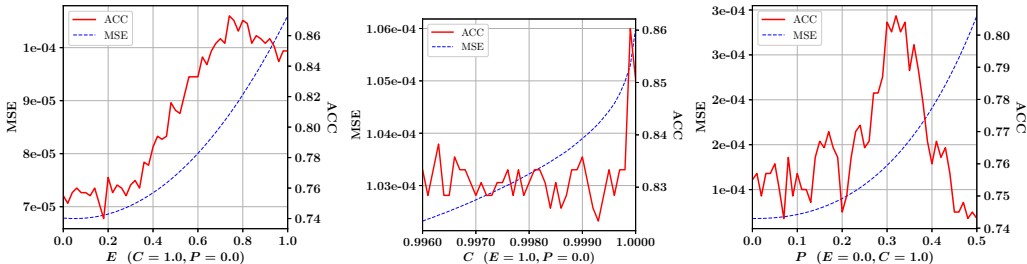

Figure 2: Test accuracy and quantization error (MSE) of fine-tuned BERT-base model on MRPC when weights are quantized into 3 bits by our proposed method while one of $E$, $C$, or $P$ varies.

### 4.3 EMPIRICAL OBSERVATIONS[1]

To verify basic operations of our proposed method, we perform post-training weighted quantization using fine-tuned models of BERT-base (Devlin et al., 2018) on MNLI and MRPC dataset within a GLUE benchmark (Wang et al., 2018). In the case of fine-tuned BERT models, we quantize all weights except those of a segment embedding layer and a classification layer which show a tiny storage footprint. For conventional or weighted Alternating quantization methods, we conduct iterative refinements of $\alpha$ and $\boldsymbol{B}$ values 20 times over which no further noticeable quantization error improvement is recognized. Given a weight matrix or tensor, $\alpha$ and $\boldsymbol{B}$ are computed for each row, independently (hence, we study row-wise quantization in this work). Due to the space limit, see Appendix for additional experimental results with various models not included in this section.

We first analyze how a simple weighted quantization scheme (with $E$=1.0, $C$=1.0, and $P$=0.0) adds distinguished features to the conventional quantization methods. Table 1 presents comparisons on quantization error (MSE), average scaling factor values, training loss, and training accuracy. Note that for Alternating weighted quantization, despite larger quantization MSE (i.e., $\sum_{i=1}^{n}(w_i - \hat{w}_i)^2$), training loss and training model accuracy are improved such that it is confirmed that minimizing $\sum_{i=1}^{n}(m_i(w_i - \hat{w}_i)^2)$ is preferred to minimizing MSE for post-training quantization. It is interesting to see that for the Greedy method, quantization MSE is reduced by weighted quantization. We conjecture that for weight distributions in DNNs, Eq. (5) is probably a better approximation compared to Eq. (1) even to minimize quantization MSE. For both Greedy and Alternating methods, scaling factors increase by weighted quantization due to the context of magnitude-based importance design.

Let us study the impact of our proposed weighted quantization on model accuracy when $E$, $C$, and $P$ can vary for fine-tuning. Figure 2 describes test accuracy and quantization error of a fine-tuned BERT-base model on MRPC when we sweep only one of $E$, $C$, or $P$ across all layers. It is clear that all hyper-parameters for importance metrics enable new search space for model accuracy that is somewhat uncorrelated to quantization error. Using various combinations of $E$, $C$, and $P$, Table 2 describes test model accuracy of fine-tuned BERT and DistilBERT models using Greedy and

---

[1]See Appendix for detailed descriptions of models and dataset selected for our experiments.

Table 2: Test score after post-training quantization with various $E$, $C$ and $P$ choices when the quantization bit is 3.

| Method | $E$ | $C$ | $P$ | BERT-base | | | DistilBERT-base | | |
|---|---|---|---|---|---|---|---|---|---|
| | | | | MRPC (Acc) | MNLI (Acc) | SQUAD (f1) | MRPC (Acc) | MNLI (Acc) | SQUAD (f1) |
| Greedy | **No Importance** | | | **78.2** | **73.4** | **56.3** | **70.0** | **58.9** | **50.4** |
| | 1 | 1.0 | 0.0 | 77.0 | **79.4** | 75.2 | 76.7 | **59.5** | 62.0 |
| | 1 | 0.99 | 0.0 | **82.1** | 79.3 | **78.9** | **78.7** | 58.5 | **64.1** |
| Alter. | **No Importance** | | | **76.0** | **81.2** | **81.9** | **75.5** | **68.2** | **77.1** |
| | 1.0 | 1.0 | 0.0 | 85.0 | 81.6 | 83.7 | 76.5 | **75.1** | 77.8 |
| | 1.0 | 0.99 | 0.0 | 83.1 | 81.3 | 83.3 | 73.0 | 71.7 | 77.1 |
| | 1.0 | 0.9999 | 0.0 | **86.0** | 81.6 | 84.3 | 74.5 | 74.3 | 76.8 |
| | 0.5 | 1.0 | 0.0 | 81.1 | 82.7 | **84.7** | 79.4 | 75.1 | **79.6** |
| | 0.2 | 1.0 | 0.0 | 76.7 | **82.8** | 83.5 | 76.2 | 71.6 | 78.1 |
| | 0.5 | 0.9999 | 0.0 | 79.9 | 82.4 | 84.1 | **78.9** | 74.4 | 79.5 |
| | 0.0 | 1.0 | 0.02 | 75.0 | 81.4 | 82.8 | 76.0 | 66.9 | 77.0 |
| | 0.0 | 1.0 | 0.05 | 76.0 | **81.6** | 82.7 | 75.5 | 66.3 | 77.3 |
| | 1.0 | 1.0 | 0.02 | 85.0 | 81.5 | 83.5 | 77.0 | 75.3 | **77.7** |
| | 1.0 | 1.0 | 0.05 | **85.2** | 81.5 | **83.7** | 77.7 | **75.6** | 76.7 |
| | 1.0 | 1.0 | 0.10 | 84.1 | 81.0 | 83.5 | **78.7** | 74.7 | 74.8 |
| Full-Precision (ours) | | | | 87.7 | 84.5 | 88.6 | 85.3 | 82.1 | 86.9 |

Alternating quantization methods. Even though numerous hyper-parameter combinations outperform Greedy and Alternating methods without importance, the best set of hyper-parameters varies for each model, and hence, an automated hyper-parameter search process is desirable. Note that in order to enable such an automated process, we need to investigate whether a set of hyper-parameters for importance searched by using training dataset is also valid for test dataset. We extensively explored hyper-parameter combinations (using $E$, $C$, $P$) with different number of refinement iterations using BERT on MRPC and MNLI and confirmed that training model accuracy achieved by our weighted quantization is highly correlated with test model accuracy such that our proposed hyper-parameters maintain a generalization capability (see Fig. 4 in Appendix).

Increasing the number of scaling factors (by decreasing the number of parameters sharing a scaling factor) enhances model accuracy despite increased memory footprint and computation overhead during inference. Our proposed weighted quantization obtains larger accuracy improvements when less number of scaling factors is utilized (see Table 7 in Appendix that compares model accuracy with different number of scaling factors when Alternating method is applied to BERT on MRPC.)

## 5 EXPERIMENTAL RESULTS

We observed that an optimal set of hyper-parameters needs to be achieved empirically. Unfortunately, optimizing $E$, $C$, and $P$ for post-training quantization to obtain the best model accuracy is challenging because 1) trained models present a variety of weight distributions and 2) hyper-parameters are correlated. As an effort to automate the hyper-parameter search process, we adopt Bayesian optimization (BO) that is implemented by a publicly available code introduced in (Nogueira, 2014–). Once we perform a rough and fast grid search for hyper-parameters (as shown in Table 2), then BO conducts fine-tuning of hyper-parameters. As a result, we achieve quick post-training quantization even when optimal hyper-parameters vary for each layer. For BO experiments, training dataset $D_t$ is used during hyper-parameter search, and then test dataset $D_v$ validates the optimization procedure (refer to Table 4). In other words, given a hyper-parameter vector denoted by $\mathbf{x} = \{E, C, P\} \in \mathbb{R}^3$, BO tries to find the optimal $\mathbf{x}^*$ to be $\arg\max_{\mathbf{x}} f(\mathbf{x}; D_t)$, where $f$ measures accuracy of the model. Then, test model accuracy is measured as $f(\mathbf{x}^*; D_v)$.

To the best of our knowledge, ***our work is the first post-training binary-coding-based quantization considering weight importance.*** As a result, we compare our results with the conventional Greedy algorithm and Alternating algorithm. We consider three different search methods for our proposed scheme: 1) **(manual search)** we investigate prearranged 16 sets of hyper-parameters for importance

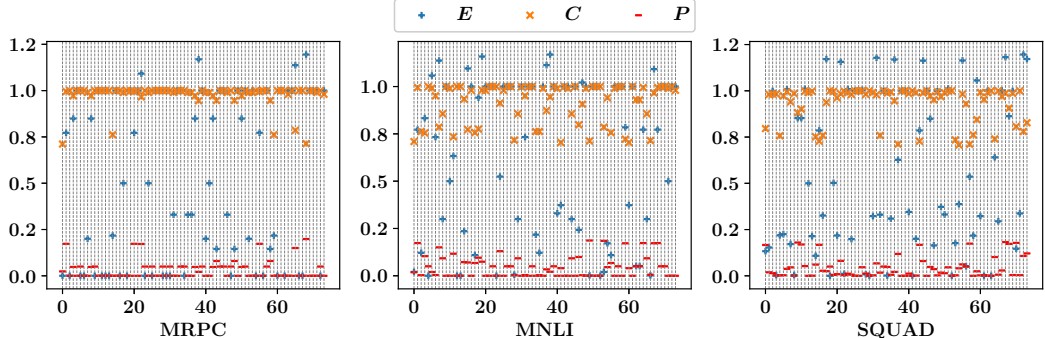

Figure 3: $E$, $C$, and $P$ values searched by layer-wise BO for BERT-base on MRPC, MNLI, and SQUAD. X-axis shows layer index and y-axis shows hyper-parameters optimized differently for each layer. BO is necessary to efficiently and quickly find such diversified $E$, $C$, and $P$ values.

Table 3: Quantization results on various language models (see Appendix for details on model descriptions). Alternating quantization scheme significantly improves test scores when combined with our proposed importance metrics (described as 'Ours') that are searched by layer-wise BO.

| Model | Dataset | Quant. Bits | Test Score | | | |
|---|---|---|---|---|---|---|
| | | | Metric | FP | Alter. | **Ours** |
| AWD-LSTM (fine-tuned) | PTB | 2 | PPL | 56.4 | 79.3 | **62.6** (-16.7) |
| | | 3 | PPL | 56.4 | 61.2 | **57.1** (-4.1) |
| BERT base | MRPC | 3 | ACC | 87.7 | 76.0 | **84.3** (+8.3) |
| | MNLI | 3 | ACC | 84.5 | 81.2 | **82.9** (+1.7) |
| | SQUAD 1.1 | 3 | F1 | 88.6 | 81.9 | **85.2** (+3.3) |
| DistilBERT base | MRPC | 3 | ACC | 85.3 | 75.5 | **83.3** (+7.8) |
| | MNLI | 3 | ACC | 82.1 | 68.2 | **79.9** (+11.7) |
| | SQUAD 1.1 | 3 | F1 | 86.9 | 77.1 | **81.2** (+4.1) |
| Longformer | SQUAD 1.1 | 4 | F1 | 89.2 | 85.8 | **86.6** (+0.8) |
| Transformer | newstest2017(en2de) | 3 | BLEU | 26.95 | 25.09 | **25.23** (+0.14) |

metric (described in Table 5), 2) **(model-wise BO)** for all layers, the same values of $E$, $C$, and $P$ are explored and applied for quantization, and 3) **(layer-wise BO)** hyper-parameters are locally searched for each layer, and the fixed before proceeding to the next layer (hence, BO for quantization is performed in layer-by-layer manner). For all model-wise or layer-wise BO for various models, the same 16 sets of hyper-parameters (given in Table 5) are explored first as initial samples. BO outperforms our manual search while layer-wise BO improves test score further as presented in Table 8 (BERT-base), 9 (DistilBERT-base), 11 (Longformer), 13 (AWD-LSTM), and 14 (Transformer NMT). Among three methods considered, layer-wise BO is the best because the optimal set of hyper-parameters turns out to be vastly different for each layer as shown in Figure 3, 5, 6, 8, and 9.

Table 3 presents the overall comparison on test scores of various language models that are quantized by conventional Alternating quantization and our proposed weighted quantization. Compared to conventional Alternating quantization (that is our baseline for post-training binary-coding-based quantization) of equal importance for each parameter, ours improves test scores for all language models in Table 3. We note that weighted quantization yields relatively different amounts of improvements on test scores depending on a given model. Even though thorough analysis of such different accuracy enhancement would entail in-depth sensitivity analysis of parameters toward test scores, pruning weights in the context of magnitude (without retraining) provides approximated correlation between importance and magnitude for a given weight distribution (see Figure 7 in Appendix). Indeed, the pruning rate of the Longformer is a lot lower compared to the other models (as shown in Figure 7) that can partly explain challenges to enhance Longformer quantization. We also note that our proposed method highly depends on the target objective function. As such, Transformers show different quantization results depending on the target selection of perplexity (PPL) or BLEU score due to somewhat low correlation between PPL and BLEU score (Appendix C.3).

We applied our weighted quantization scheme to ResNet models on CIFAR-10 and ImageNet (refer to Appendix D) for which model accuracy is also significantly enhanced similar to language models.

## 6 CONCLUSIONS AND FUTURE WORK

In this paper, we propose a weighted quantization framework employing importance metrics that are useful when each parameter shows a different sensitivity toward a loss function change. For the binary-coding-based quantization that is our choice for language models because of existing efficient kernel designs (e.g., `BiQGEMM`) and high compression ratio, we extract modified Greedy method and Alternating method assuming that each importance value is represented as a real number between 0 and 1. Using various DNN models, we demonstrate that a magnitude-based importance metric is effective for post-training quantization in the form of the binary codes. To fine-tune model accuracy, we also propose three hyper-parameters that need to be empirically investigated since an optimal set of hyper-parameters varies depending on each model design. As such, we suggest Bayesian optimization as an effective technique to automate hyper-parameter search process. Our proposed hyper-parameters can be independently optimized to each layer to further improve compression ratio and/or model accuracy. It would be interesting to study additional hyper-parameters effective for post-training quantization. Since our weighted quantization framework is general (rather than depending on a particular approximation such as the Hessian), if proper importance metrics are found, our proposed quantization techniques can be extended to a quantization-aware training method.

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

# A    MODELS AND DATASETS

## A.1    LSTM MODELS

**1-Layer LSTM model (in Fig. 1) (Zaremba et al., 2014)**: One layer LSTM model with 300 hidden states [2] is used to predict PTB dataset. We compress the a LSTM layer and an embedding layer of the pre-trained language model to draw the Figure 1. A scaling factor is extracted for each raw, e.g., for the (10000, 300) embedding layer, there exist 10,000 scaling factors as a result of quantization.

**AWD-LSTM (Merity et al., 2017)**: We use a 3-layer AWD-LSTM model. The embedding size is 400 and the hidden vector size within LSTM layers is 1550. We train the original model during 500 epochs and then fine-tune during additional 300 epochs [3]. We compress both models including embedding and softmax layers by our post-training quantization method.

**Dataset**: To evaluate our quantization method, we use the `test` dataset of Penn Treebank (PTB) dataset (Marcus et al., 1993) for the LSTM language models. To find the quantization parameters with BO, we use `valid` dataset.

## A.2    HUGGINGFACE LANGUAGE MODELS

To evaluate recently developed language models, we utilize the `transformers` library (PyTorch version) developed by `huggingface` (Wolf et al., 2019). For all bert-based models, the last classification layer and the sequence embedding layer are not quantized because the sizes of weights are relatively smaller than other weights.

**BERT (Devlin et al., 2018) / DistilBERT (Sanh et al., 2019)**: We fine-tune the pre-trained BERT-base and DistilBERT-base models to evaluate our method. BERT-base model consists of 12 encoder blocks with 768 hidden size and DistilBERT-base model consists of 6 encoder blocks with 768 hidden size. We follow fine-tuning recipes found in the `transformers` repository [4]. For the MRPC and MNLI tasks, the initial learning rate is 2e-5 and the training epoch is 3. For the SQUAD task, the initial learning rate is 3e-5 and the training epoch is 2.

**Longformer (Beltagy et al., 2020)**: To evaluate the Longformer model, we choose the pre-trained longformer-base model that has 12 blocks of Longformer encoder with 768 hidden size. We fine-tune the pre-trained model[5] for SQUAD v1.1 dataset with the same recipe.

**Dataset**: We use three language tasks: MRPC, MNLI and SQUAD(v1.1). To search quantization parameters ($E$, $C$ and $P$), we use a randomly sampled fraction of `train` dataset when the evaluation time is too time-consuming. For the MRPC task, we use the whole `train` dataset because the `train` dataset is small enough to use. For the MNLI task, we use only 10% of `train` dataset. For the SQUAD task, we use only 6.7% of `train` dataset for bayesian optimization while `dev` dataset is used for testing because there is no published `test` dataset.

## A.3    OPENNMT TRANSFORMER

We use the pre-trained transformer model for Neural Machine Translation task (Klein et al., 2018) [6]. The transformer model consists of 6 encoder blocks, 6 decoder blocks and embedding layers. Note that the embedding layers are not shared, e.g. embedding weights are not tied. The vocabulary size is 50k and the size of the hidden vector is 512.

**Dataset**: We evaluate the pre-trained model in a translation direction: English to German (en2de). We use `valid` dataset in newstest2017 for BO processes and `test` dataset for the test evaluation. All datasets are pre-processed by SentencePiece (Kudo & Richardson, 2018). All the translation scores are BLEU scores by `sacrebleu` script (Post, 2018) as the beam size is 1.

---

[2]Available at https://github.com/pytorch/examples/tree/master/word_language_model

[3]The detailed parameters are described in https://github.com/salesforce/awd-lstm-lm

[4]https://github.com/huggingface/transformers

[5]Avaliable at https://huggingface.co/allenai/longformer-base-4096

[6]Available at https://opennmt.net/Models-py/

## A.4 RESNET FOR IMAGE CLASSIFICATION

We conduct experiments using ResNet32 on CIFAR10 (Krizhevsky, 2009) and ResNet18 (He et al., 2016) on ImageNet (Russakovsky et al., 2015). For convolution tensors, $\alpha$ and B are computed for each channel. We maintain full-precision on the first and last layers of ResNet models because those layers are very small while a lot of quantization bits are required (McDonnell, 2018).

For ResNet-18, we use ImageNet1K[7] training dataset, which is a small subset of ImageNet dataset, for fast hyper-parameter search, while test accuracy is still measured by the entire ImageNet test dataset. To obtain the same accuracy for the same set of hyper-parameters, the training dataset is not randomly manipulated (e.g., by cropping and flipping).

## B    BAYESIAN OPTIMIZATION FOR WEIGHTED QUANTIZATION

BO is one of automated machine learning (AutoML) techniques to search optimal hyper-parameters for target networks. Given *a black box* function $f$, BO aims to find the optimal $\mathbf{x}^*$ to be $\arg\max_{\mathbf{x}} f(\mathbf{x})$. Suppose observations are described as $\mathbf{y} = [f(\mathbf{x}_1), f(\mathbf{x}_2), \ldots, f(\mathbf{x}_n)]^T$ and $y_*$ is an output of any unobserved $\mathbf{x}_*$, then under the assumption that $f(\mathbf{x})$ is drawn from *Gaussian Process*, the distribution of $y_*|\mathbf{y}$ follows $\mathcal{N}(K_* K^{-1}\mathbf{y}, K_{**} - K_* K^{-1} K_*^T)$, where

$$K = \begin{bmatrix} k(\mathbf{x}_1, \mathbf{x}_1) & \cdots & k(\mathbf{x}_1, \mathbf{x}_n) \\ \vdots & \ddots & \vdots \\ k(\mathbf{x}_1, \mathbf{x}_1) & \cdots & k(\mathbf{x}_1, \mathbf{x}_n) \end{bmatrix}, K_* = [k(\mathbf{x}_*, \mathbf{x}_1) \cdots k(\mathbf{x}_*, \mathbf{x}_n)], \text{ and } K_{**} = k(\mathbf{x}_*, \mathbf{x}_*). \quad (10)$$

The kernel function $k(\mathbf{x}, \mathbf{x}')$ is one of the hyper-parameter for BO and measures a similarity between $\mathbf{x}$ and $\mathbf{x}'$ (i.e., the output is high when they are close). There are various kernel functions (Rasmussen & Williams, 2006), and we use squared-exponential kernel that is one of the popular choices for regression (Ebden, 2015). To identify which of the unobserved data to be taken as $\mathbf{x}_{n+1}$, the acquisition function needs to be specified. In general, the expected improvement acquisition function $a_{EI}$ (Lizotte, 2008) (see Eq. (11)) is most commonly used and is selected for our experiments.

$$a_{EI}(\mathbf{x}_*|\mathbf{y}) = (Z\Phi(Z) + \phi(Z))\sigma(\mathbf{x}_*), \quad (11)$$

where

$$Z = \begin{cases} \frac{\mu(\mathbf{x}_*) - f(\mathbf{x}^+) - \xi}{\sigma(\mathbf{x}_*)} & \text{if } \sigma(\mathbf{x}_*) > 0 \\ 0 & \text{if } \sigma(\mathbf{x}_*) = 0 \end{cases} \text{ and } f(\mathbf{x}^+) = \max_{1 \leq i \leq n} f(\mathbf{x}). \quad (12)$$

The parameter $\xi$ is the trade-off factor between *exploitation* and *exploration*. In our experiments, we set $\xi$ to 0.2, which implies that *exploitation* has more influence on determining $\mathbf{x}_{n+1}$. After computing $a_{EI}$ for unobserved random sampled data $\mathbf{x}_*$s, $\mathbf{x}_{n+1}$ is computed as $\arg\max_{\mathbf{x}_*} a_{EI}(\mathbf{x}_*|\mathbf{y})$. Further details of BO can be found in (Brochu et al., 2010; Nogueira, 2014–; Snoek et al., 2012).

---

[7]Available at: https://s3.amazonaws.com/pytorch-tutorial-assets/imagenet_1k.zip

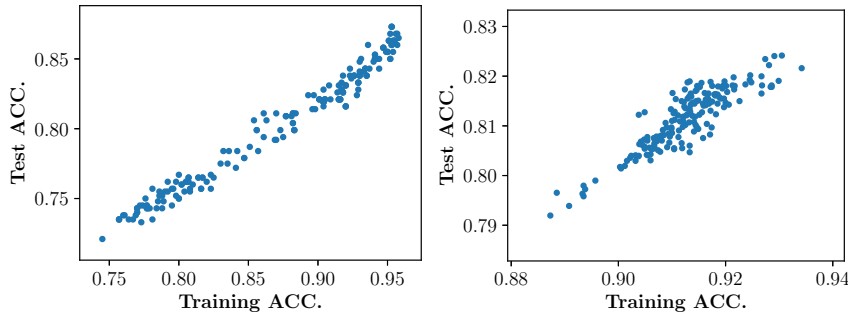

Figure 4: Relationship of training accuracy achieved by weighted quantization and test accuracy using BERT on MRPC (LEFT) and BERT on MNLI (RIGHT).

Table 4: Dataset usages and maximum iterations for Bayesian optimization. In the case of large training dataset (such as MNLI and SQUAD v1.1), we use samples.

| | Dataset | | Maximum iterations for BO | |
|---|---|---|---|---|
| | BO Process | Test Evaluation | Model-wise | Layer-wise |
| Penn Treebank | valid | test | 1000 | 250 |
| MRPC | train | test | 1000 | 50 |
| MNLI | 10% of train | test | 1000 | 50 |
| SQUAD v1.1 | 6.7% of train | dev | 1000 | 50 |
| newstest2017 | valid (detokenized) | test (detokenized) | 1000 | 30 |
| CIFAR10 | train | test | 2000 | 100 |
| ImageNet | train of ImageNet1K | test | 2000 | 200 |

## C RESULTS ON LANGUAGE MODELS

Table 5: 16 sets of hyper-parameters selected for our manual search of importance metric.

| | | Set 1 | Set 2 | Set 3 | Set 4 | Set 5 | Set 6 | Set 7 | Set 8 |
|---|---|---|---|---|---|---|---|---|---|
| $P$=0.0 (default) | $E$ | 1.0 | 1.0 | 1.0 | 1.0 | 0.5 | 0.2 | 0.5 | 0.5 |
| | $C$ | 1 | 0.99 | 0.999 | 0.9999 | 1.0 | 1.0 | 0.999 | 0.9999 |
| | | Set 9 | Set 10 | Set 11 | Set 12 | Set 13 | Set 14 | Set 15 | Set 16 |
| $C$=1.0 (default) | $E$ | 0.0 | 0.0 | 0.0 | 0.0 | 1.0 | 1.0 | 1.0 | 1.0 |
| | $P$ | 0.02 | 0.05 | 0.1 | 0.2 | 0.02 | 0.05 | 0.1 | 0.2 |

### C.1 FINE-TUNED BERT, DISTILBERT AND LONGFORMER

Table 6: Test score after post-training quantization with various $E$, $C$ and $P$ choices when the quantization bit is 4.

| Method | $E$ | $C$ | $P$ | BERT-base MRPC (Acc) | MNLI (Acc) | SQUAD (f1) | DistilBERT-base MRPC (Acc) | MNLI (Acc) | SQUAD (f1) |
|---|---|---|---|---|---|---|---|---|---|
| Greedy | **No Importance** | | | **76.2** | **78.1** | **70.7** | **77.7** | **64.7** | **66.6** |
| | 1 | 1.0 | 0.0 | 72.3 | 80.1 | 79.0 | 73.0 | 73.4 | 80.7 |
| | 1 | 0.99 | 0.0 | **80.1** | **81.0** | **81.8** | **76.0** | **75.2** | **81.2** |
| Alter. | **No Importance** | | | 78.4 | 82.8 | 86.2 | 77.0 | 77.0 | 83.4 |
| | 1.0 | 1.0 | 0.0 | 86.0 | 83.8 | 88.0 | **84.1** | 81.4 | **85.9** |
| | 1.0 | 0.99 | 0.0 | 84.3 | 83.5 | 87.9 | 82.3 | 80.9 | 85.2 |
| | 1.0 | 0.999 | 0.0 | 86.3 | 83.6 | 87.8 | 83.3 | **81.4** | 85.7 |
| | 1.0 | 0.9999 | 0.0 | **86.5** | 83.7 | 88.0 | 84.1 | 81.6 | 85.7 |
| | 0.5 | 1.0 | 0.0 | 85.0 | **83.9** | **88.1** | 83.3 | 80.9 | 85.2 |
| | 0.2 | 1.0 | 0.0 | 82.6 | 83.3 | 87.0 | 81.6 | 78.9 | 84.4 |
| | 0.5 | 0.999 | 0.0 | 85.0 | 83.6 | 87.8 | 82.1 | 79.7 | 84.9 |
| | 0.5 | 0.9999 | 0.0 | 85.5 | 83.5 | 87.9 | 82.4 | 80.2 | 85.1 |
| | 0.0 | 1.0 | 0.02 | 78.9 | 82.8 | 86.4 | 78.4 | 77.1 | 83.8 |
| | 0.0 | 1.0 | 0.05 | 79.9 | 82.9 | 86.3 | 79.7 | 77.4 | 83.8 |
| | 0.0 | 1.0 | 0.1 | 80.4 | 83.0 | 86.4 | 79.4 | 77.2 | 83.6 |
| | 0.0 | 1.0 | 0.2 | 81.6 | 82.8 | 86.2 | 79.9 | 77.4 | 83.5 |
| | 1.0 | 1.0 | 0.02 | **86.5** | **83.9** | 88.0 | **84.6** | 81.6 | **85.8** |
| | 1.0 | 1.0 | 0.05 | 86.3 | 83.7 | 88.0 | 84.1 | **81.7** | 85.6 |
| | 1.0 | 1.0 | 0.1 | 85.8 | 83.7 | 87.9 | 84.1 | 81.4 | 85.4 |
| | 1.0 | 1.0 | 0.2 | 84.3 | 83.7 | 87.5 | 83.8 | 81.3 | 84.9 |
| Full-Precision (ours) | | | | 87.7 | 84.5 | 88.6 | 85.3 | 82.1 | 86.9 |

Table 7: Model accuracy of BERT on MRPC quantized into 3 bits per weight by Alternating method with different number of parameters sharing a scaling factor.

| $E$ | $C$ | $P$ | # of parameters sharing one $\alpha$ 48 | 96 | 192 | 384 | 768 |
|---|---|---|---|---|---|---|---|
| **No Importance** | | | **85.29** | **85.05** | **79.41** | **76.96** | **75.74** |
| 1 | 1 | 0 | 85.05 | 85.29 | 85.29 | 84.31 | **84.80** |
| 1 | 0.95 | 0 | 85.05 | 84.31 | 85.54 | 83.09 | 79.90 |
| 1 | 0.99 | 0 | 85.05 | 84.31 | 85.05 | 82.84 | 81.62 |
| 1 | 0.999 | 0 | 84.80 | 84.56 | **85.78** | 84.07 | 83.82 |
| 1 | 0.9999 | 0 | 84.56 | 84.07 | 85.05 | **84.56** | 82.35 |
| 0.5 | 1 | 0 | 85.78 | **86.03** | 85.29 | 83.09 | 79.90 |
| 0.2 | 1 | 0 | **86.03** | 85.05 | 82.35 | 80.64 | 75.25 |
| 0 | 1 | 0.05 | 84.31 | 84.80 | 75.25 | 75.98 | 76.00 |
| 1 | 1 | 0.05 | 84.07 | 85.05 | 84.56 | 83.84 | 84.12 |

Table 8: Hyper-parameter search results of BERT-base (quantized into 3 bits per weight) using manual search, model-wise BO, and layer-wise BO.

| | Hyper-parameter search method | train ACC. | test ACC. | Parameters($E$,$C$,$P$) |
|---|---|---|---|---|
| MRPC | Manual (by Table 5) | - | 86.0 | (1.0, 0.9999, 0.0) |
| | Model-wise BO | 93.8 | 86.5 | (0.693, 0.999, 0.045) |
| | Layer-wise BO | 94.5 | 84.3 | Fig. 3 |
| MNLI | Manual (by Table 5) | - | 82.8 | (0.2, 1.0, 0.0) |
| | Model-wise BO | 93.7 | 82.4 | (0.534, 0.9999, 0.002) |
| | Layer-wise BO | 94.5 | 82.9 | Fig. 3 |
| SQUAD | Manual (by Table 5) | - | 84.7 | (0.5, 1.0, 0.0) |
| | Model-wise BO | 90.0 | 84.8 | (0.560, 1.0, 0.0) |
| | Layer-wise BO | 90.7 | 85.2 | Fig. 3 |

Table 9: Hyper-parameter search results of DistilBERT-base (quantized into 3 bits per weight) using manual search, model-wise BO, and layer-wise BO.

|  | Hyper-parameter search method | train ACC. | test ACC. | Parameters($E$,$C$,$P$) |
|---|---|---|---|---|
| MRPC | Manual (by Table 5) | - | 78.9 | (0.5, 0.9999, 1.0) |
|  | Model-wise BO | 85.2 | 79.7 | (0.679, 1.0, 0.0) |
|  | Layer-wise BO | 90.3 | 83.3 | Fig. 5 |
| MNLI | Manual (by Table 5) | - | 75.6 | (1.0, 1.0, 0.05) |
|  | Model-wise BO | 80.7 | 75.4 | (0.687, 1.0, 0.048) |
|  | Layer-wise BO | 86.9 | 79.9 | Fig. 5 |
| SQUAD | Manual (by Table 5) | - | 79.6 | (0.5, 1.0, 0.0) |
|  | Model-wise BO | 86.6 | 79.6 | (0.430, 1.0, 0.0) |
|  | Layer-wise BO | 88.4 | 81.2 | Fig. 5 |

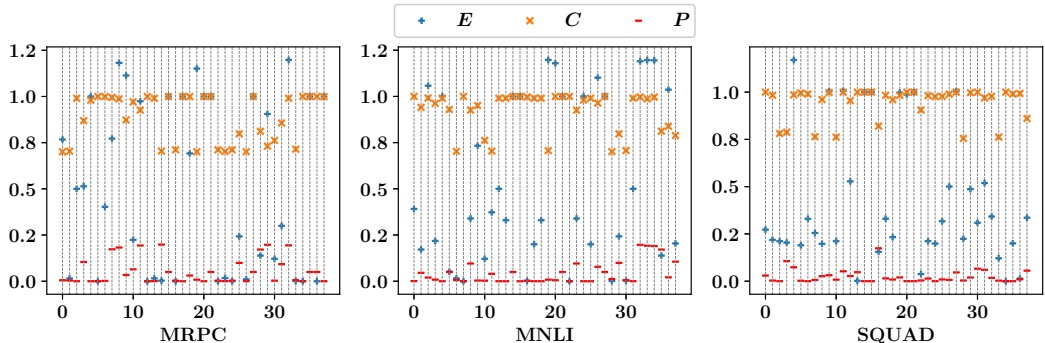

Figure 5: $E$, $C$, and $P$ values searched by layer-wise BO for DistilBERT-base on MRPC, MNLI, and SQUAD. X-axis shows layer index and y-axis shows hyper-parameters optimized differently for each layer.

Table 10: F1 scores of Longformer on SQUAD v1.1 after post-training quantization (4 bits per weight) with various $E$, $C$ and $P$ choices.

| Method | $E$ | $C$ | $P$ | 4bit | 5bit |
|---|---|---|---|---|---|
|  | **No Importance** | | | **85.8** | **88.0** |
|  | 1.0 | 1.0 | 0.0 | 83.5 | 88.2 |
|  | 1.0 | 0.99 | 0.0 | 83.8 | 88.3 |
|  | 0.5 | 1.0 | 0.0 | 85.2 | 88.3 |
|  | 0.2 | 1.0 | 0.0 | 85.9 | 87.8 |
|  | 0.5 | 0.999 | 0.0 | 85.4 | 88.3 |
| Alter. | 0.5 | 0.9999 | 0.0 | 85.4 | **88.4** |
|  | 0.0 | 1.0 | 0.02 | **86.0** | 88.1 |
|  | 0.0 | 1.0 | 0.05 | **86.0** | 87.7 |
|  | 0.0 | 1.0 | 0.1 | 85.4 | 87.7 |
|  | 1.0 | 1.0 | 0.02 | 83.6 | 88.0 |
|  | 1.0 | 1.0 | 0.05 | 83.0 | 88.1 |
|  | 1.0 | 1.0 | 0.1 | 81.4 | 88.1 |
| Full-Precision | | | | 89.2 | |

Table 11: Hyper-parameter search results of Longformer using manual search, model-wise BO, and layer-wise BO.

|  | Hyper-parameter search method | val ACC. | test ACC. | Parameters($E$,$C$,$P$) |
|---|---|---|---|---|
| 4-bit | Manual (by Table 5) | - | 86.0 | (0.0, 1.0, 0.05) |
|  | Model-wise BO | 88.9 | 86.5 | (0.11, 1.0, 0.0) |
|  | Layer-wise BO | 89.6 | 86.6 | Fig. 6 |

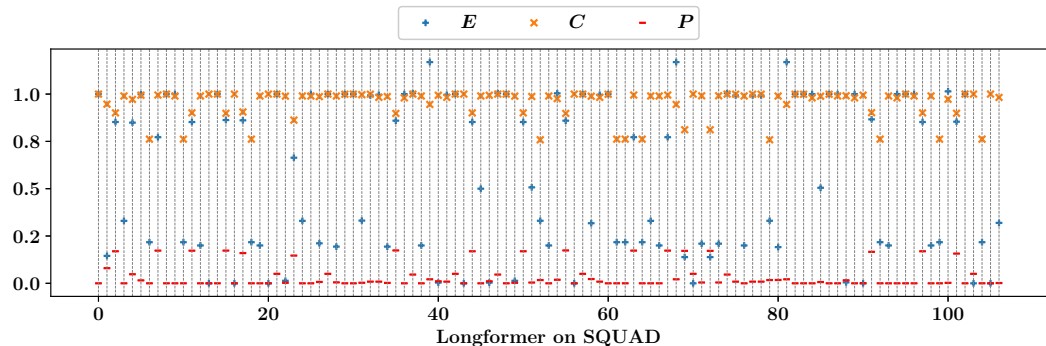

Figure 6: $E$, $C$, and $P$ values searched by layer-wise BO for Longformer on SQUAD v1.1. X-axis shows layer index and y-axis shows hyper-parameters optimized differently for each layer.

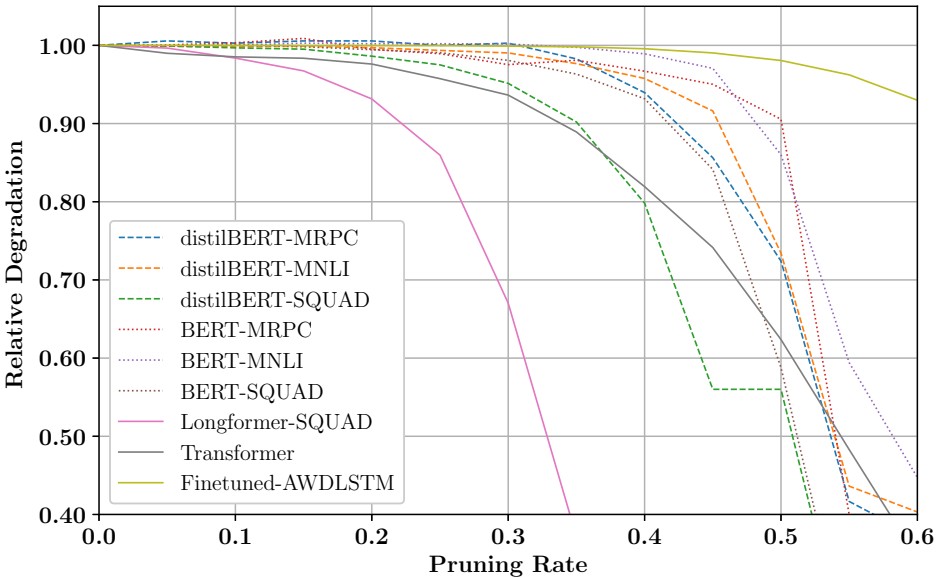

Figure 7: Test score degradation by (post-training) pruning weights (based on the magnitude) using various pre-trained language models. Weights of a layer are pruned by the same target pruning rate. For the same pruning rate, the Longformer presents sharper score degradation that partly explains the difficulty of improving test scores by our proposed weighted quantization method compared to the conventional Alternating quantization.

## C.2 AWD-LSTM ON PTB DATASET

Table 12: Perplexity of AWD-LSTM model on PTB test dataset after post-training quantization with various $E$, $C$ and $P$ choices.

| Method | $E$ | $C$ | $P$ | Trained | | | Fine-tuned | | |
|---|---|---|---|---|---|---|---|---|---|
| | | | | 2bit | 3bit | 4bit | 2bit | 3bit | 4bit |
| | **No Importance** | | | **93.25** | **66.57** | **61.84** | **79.32** | **61.21** | **58.11** |
| | 1.0 | 1.0 | 0.0 | **71.01** | 62.16 | 59.97 | 68.38 | 59.19 | **57.14** |
| | 1.0 | 0.99 | 0.0 | 68.47 | 62.29 | 60.46 | 65.66 | 59.14 | 57.23 |
| | 1.0 | 0.999 | 0.0 | 69.13 | **62.01** | 60.04 | 66.91 | 59.08 | 57.19 |
| | 1.0 | 0.9999 | 0.0 | 69.95 | 62.10 | 59.99 | 67.73 | 59.16 | **57.14** |
| | 0.5 | 1.0 | 0.0 | 72.38 | 63.04 | 60.50 | 65.04 | 59.05 | 57.65 |
| | 0.2 | 1.0 | 0.0 | 81.96 | 64.76 | 61.14 | 71.28 | 60.07 | 57.82 |
| | 0.5 | 0.999 | 0.0 | 72.52 | 63.07 | 60.58 | 65.05 | 59.05 | 57.68 |
| Alter. | 0.5 | 0.9999 | 0.0 | 72.32 | 63.05 | 60.5 | **64.99** | **59.03** | 57.59 |
| | 0.0 | 1.0 | 0.02 | 91.84 | 66.42 | 61.86 | 78.32 | 61.13 | 58.14 |
| | 0.0 | 1.0 | 0.05 | 89.70 | 66.22 | 61.85 | 76.84 | 60.95 | 58.16 |
| | 0.0 | 1.0 | 0.1 | 86.41 | 65.78 | 61.76 | 74.70 | 60.66 | 58.20 |
| | 0.0 | 1.0 | 0.2 | 81.01 | 64.67 | 61.45 | 71.80 | 59.91 | 57.96 |
| | 1.0 | 1.0 | 0.02 | 114.22 | 62.21 | **59.96** | 68.75 | 59.27 | **57.14** |
| | 1.0 | 1.0 | 0.05 | 114.41 | 62.24 | **59.96** | 69.57 | 59.31 | **57.14** |
| | 1.0 | 1.0 | 0.1 | 114.91 | 62.32 | 59.98 | 111.77 | 59.46 | **57.14** |
| | 1.0 | 1.0 | 0.2 | 116.81 | 62.46 | 60.1 | 114.6 | 60.02 | 57.30 |
| Full-Precision | | | | | 59.13 | | | 56.43 | |

Table 13: Hyper-parameter search results of fine-tuned AWD-LSTM (quantized into 2/3/4 bits per weight) using manual search, model-wise BO, and layer-wise BO.

| | Hyper-parameter search method | val ACC. | test ACC. | Parameters($E$,$C$,$P$) |
|---|---|---|---|---|
| 2-bit | Manual (by Table 5) | - | 64.99 | (0.5, 0.9999, 0.0) |
| | Model-wise BO | 67.76 | 65.2 | (0.9934, 0.8454, 0.0) |
| | Layer-wise BO | 64.80 | 62.63 | Fig. 8 |
| 3-bit | Manual (by Table 5) | - | 59.03 | (0.5, 0.9999, 0.0) |
| | Model-wise BO | 61.34 | 58.97 | (0.9643, 0.9989, 0.0) |
| | Layer-wise BO | 60.52 | 58.21 | Fig. 8 |
| 4-bit | Manual (by Table 5) | - | 57.14 | (1.0, 1.0, 0.02) |
| | Model-wise BO | 59.55 | 57.19 | (1.0148, 0.9810, 0.029) |
| | Layer-wise BO | 59.40 | 57.11 | Fig. 8 |

## C.3 TRANSFORMER

Table 14: Quantizing Transformer by using 3 bits per weight with different quantization schemes and the metric to be optimized by BO.

| Method | Target | valid PPL | valid BLUE | test PPL | test BLEU |
|---|---|---|---|---|---|
| Pre-trained | - | 1.00373 | 25.85 | 1.00398 | 26.95 |
| Alternating (No Importance) | MSE | **1.00646** | **24.17** | **1.00678** | **25.09** |
| Model-wise BO | PPL | **1.00339** | 19.20 | **1.00368** | 20.38 |
| Layer-wise BO | PPL | **1.00283** | 23.40 | **1.00291** | 24.67 |
| Model-wise BO | BLEU | 1.00651 | **24.24** | 1.00676 | **25.05** |
| Layer-wise BO | BLEU | 1.00494 | **24.30** | 1.00498 | **25.23** |

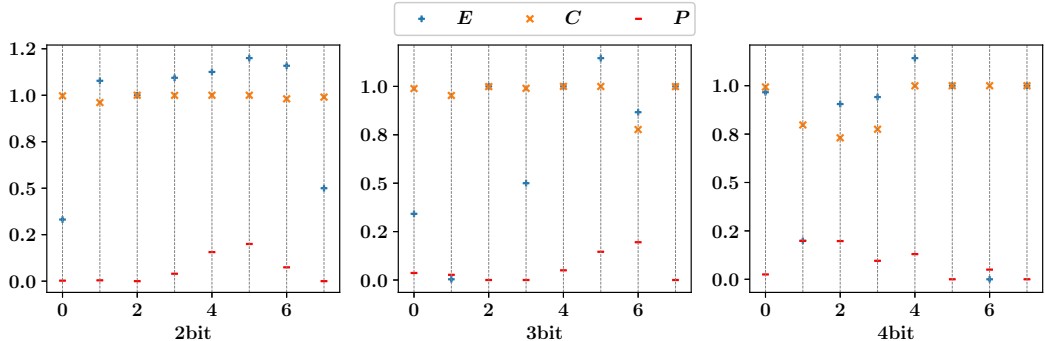

Figure 8: $E$, $C$, and $P$ values searched by layer-wise BO for fine-tuned AWD-LSTM model. X-axis shows layer index and y-axis shows hyper-parameters optimized differently for each layer.

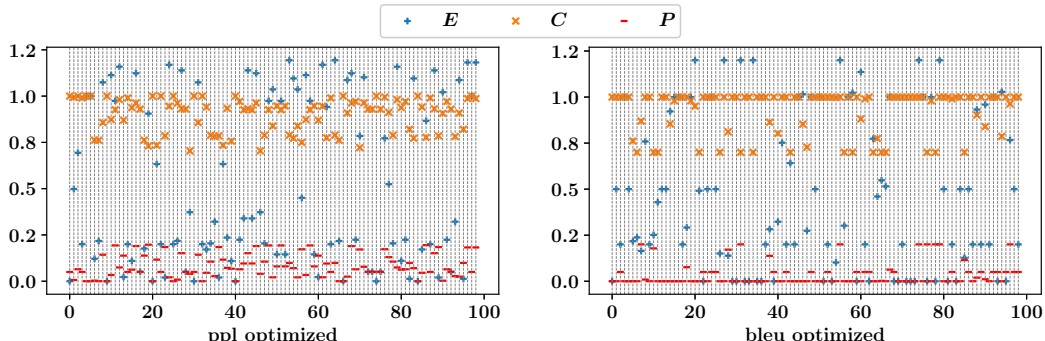

Figure 9: $E$, $C$, and $P$ values searched by layer-wise BO for Transformer. BO is performed to optimize PPL (Left) or BLEU (Right). X-axis shows layer index and y-axis shows hyper-parameters optimized differently for each layer.

# D RESULTS ON RESNET MODELS

Table 15: Post-training quantization comparison on quantization MSE, average scaling factor values, training loss, and training model accuracy. For importance metrics, $E=1.0$ is used while $P$ and $C$ are not considered.

| Model | Method | Quant. Err. (MSE) | Average Scaling Factors | Training Loss | Training Accuracy (%) |
|---|---|---|---|---|---|
| ResNet-32 (CIFAR-10) | Greedy (No Importance) | 4.35E-04 | 0.035 | 0.783 | 12.54 |
| | Greedy ($E=1.0$) | 3.02e-04 | 0.051 | 0.693 | 84.12 |
| | Alter. (No Importance) | 2.27e-04 | 0.051 | 0.324 | 90.72 |
| | Alter. ($E=1.0$) | 3.79e-04 | 0.062 | 0.309 | 91.68 |
| ResNet-18 (ImageNet) | Greedy (REF) | 4.98E-05 | 0.010 | 7.075 | 0.20 |
| | Greedy ($E=1.0$) | 3.47e-05 | 0.015 | 4.853 | 12.87 |
| | Alter. (REF) | 2.39e-05 | 0.016 | 2.419 | 47.62 |
| | Alter. ($E=1.0$) | 4.43e-05 | 0.023 | 3.033 | 37.01 |
| | Alter. ($E=2.0$) | 2.89e-05 | 0.012 | 1.674 | 59.94 |
| | Alter. ($E=3.0$) | 2.62e-05 | 0.019 | 1.992 | 54.19 |
| | Alter. ($E=5.0$) | 2.48e-05 | 0.018 | 2.024 | 54.16 |
| | Alter. ($E=7.0$) | 2.42e-05 | 0.017 | 2.282 | 49.53 |

Table 16: Model accuracy(%) on test dataset after post-training quantization with various $E$ and $C$ choices. $q$ is the number of quantization bits.

| Method | $E$ | $C$ | $P$ | ResNet-32 (CIFAR-10) $q=3$ | ResNet-32 (CIFAR-10) $q=4$ | ResNet-18 (ImageNet) $q=3$ | ResNet-18 (ImageNet) $q=4$ |
|---|---|---|---|---|---|---|---|
| Greedy | **No Importance** | | | **12.7** | **24.4** | **0.2** | **0.4** |
| | 1 | 1.0 | 0.0 | **79.9** | **84.2** | 12.0 | **44.2** |
| | 1 | 0.99 | 0.0 | 71.5 | 81.8 | **14.0** | 42.6 |
| Alter. | **No Importance** | | | **84.9** | **91.2** | **43.3** | **60.1** |
| | 1 | 1.0 | 0.0 | 85.5 | 91.7 | 32.8 | 61.1 |
| | 1 | 0.95 | 0.0 | 85.9 | 91.7 | 44.5 | 61.5 |
| | 1 | 0.99 | 0.0 | 87.2 | 91.5 | 35.8 | 60.9 |
| | 1 | 0.999 | 0.0 | 85.5 | 91.6 | 36.2 | **64.5** |
| | 1 | 0.9999 | 0.0 | 85.5 | 91.7 | 32.0 | 60.6 |
| | 0.5 | 1.0 | 0.0 | 87.6 | **91.9** | 53.4 | 61.8 |
| | 0.33 | 1.0 | 0.0 | 87.6 | 91.4 | 48.4 | 63.0 |
| | 0.2 | 1.0 | 0.0 | 86.5 | 91.6 | 48.6 | 63.5 |
| | 0.5 | 0.99 | 0.0 | 86.7 | 91.6 | 50.6 | 63.4 |
| | 0.5 | 0.999 | 0.0 | 87.3 | 91.9 | **53.4** | 61.5 |
| | 0.0 | 1.0 | 0.02 | 84.6 | 90.6 | 44.3 | 60.6 |
| | 0.0 | 1.0 | 0.05 | 84.2 | 90.8 | 43.5 | 59.3 |
| | 0.0 | 1.0 | 0.10 | 85.7 | 90.9 | 33.8 | 57.9 |
| | 0.0 | 1.0 | 0.20 | 86.8 | 90.7 | 41.4 | 62.0 |
| | 1.0 | 1.0 | 0.02 | 85.6 | 91.8 | 32.5 | 59.8 |
| | 1.0 | 1.0 | 0.05 | 85.9 | 91.7 | 31.5 | 59.6 |
| | 1.0 | 1.0 | 0.10 | **88.3** | 91.8 | 31.2 | 59.4 |
| | 1.0 | 1.0 | 0.20 | 86.0 | 91.7 | 29.3 | 61.6 |
| Full-Precision | | | | 92.8 | | 69.2 | |

Table 17: The optimal hyper-parameters searched by Bayesian optimization when Alternating quantization method is utilized and $q$ is the number of quantization bits.

| | $q$ | Alternating Test Acc.(%) | Proposed weighted quantization $E$ [0.2, 1.2] | $C$ [0.7, 1.0] | $P$ [0.0, 0.3] | Train Acc.(%) | Test Acc.(%) |
|---|---|---|---|---|---|---|---|
| ResNet-18 (ImageNet) | 3 | 43.3 | 0.4490 | 1.0000 | 0.0121 | 62.3 | **54.0** |
| | | | Layer-wise (Fig. 10) | | | 66.4 | **56.5** |
| | 4 | 60.1 | 0.5253 | 0.9501 | 0.1780 | 73.0 | **64.2** |
| | | | Layer-wise (Fig. 10) | | | 74.8 | **65.3** |
| | 5 | 66.8 | 0.5311 | 0.9783 | 0.0085 | 76.3 | **68.1** |
| | | | Layer-wise (Fig. 10) | | | 77.4 | **68.5** |
| ResNet-32 (CIFAR-10) | 3 | 84.9 | 0.4482 | 1.0000 | 0.1846 | 96.96 | **89.36** |
| | | | Layer-wise | | | 98.33 | **90.60** |
| | 4 | 91.2 | 1.0500 | 1.0000 | 0.2290 | 99.80 | **92.01** |
| | | | Layer-wise | | | 99.85 | **92.32** |

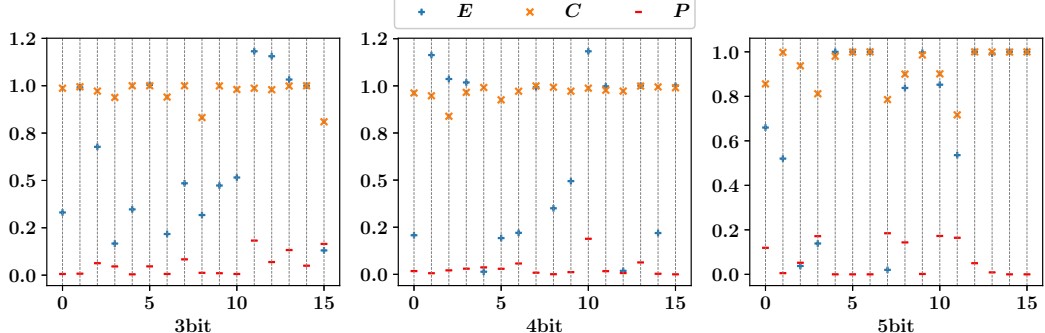

Figure 10: $E$, $C$, and $P$ values searched by layer-wise BO for ResNet18 on ImageNet. X-axis shows layer index and y-axis shows hyper-parameters optimized differently for each layer.

