# OpenReview forum: "Post-Training Weighted Quantization of Neural Networks for Language Models"
_ICLR.cc/2021/Conference — Reject_

### Official Review · AnonReviewer3 · 2020-10-17
**Less Innovative Post-Training Quantization Method**

**Rating:** 5
**Confidence:** 4

**Review:**

Based on a previous classic binary coding scheme, this paper proposed to introduce a modification $m_i$ on the binarization scaling factor $\alpha$, by considering the weight magnitude. It further use 3 hyperparamters to refine $m_i$ by constraining its upper/lower bound and exponent. Besides,  this work spent lengthy content to describe how to determine the hyperparamters.

This paper contains the following drawbacks:
1. The proposed post-training quantization method has no connection to language model. All we can say is that: Post-training weighted quantization of neural network *in* (or applied in) language models. As author also mentioned that this method also works well in image tasks.
2. There is a confusion on experimental setting: The quantization should be applied to the *fine-tuned* model. But it seems that author didn't pay attention to the disambiguation of pre-trained and fine-tuned, as I am quite lost in Sec.4.3 where author mention that "perform post-training ... using *pre-trained* models of BERT-base ... on MNLI and MRPC." And didn't emphasize that the quantization is conducted on the fine-tuned models by MNLI/MRPC.
3. Author spent much effort on how to determine the hyperparameters, which is also one defect of the method: it requires exquisite tuning on hyperparameters, which are sensitive as shown in Table 2.

Question:
1. How does the work deal with parameters in batch normalization layers?

---

> ### Author Response · Authors · 2020-11-13
> **Response to AnonReviewer3**
>
> First of all, we appreciate your effort and time to review the manuscript. We address your concerns in detail below.
>
> R1: The proposed post-training quantization method has no connection to language model.
>
> A1: We acknowledge that the proposed post-training quantization can also be applied to image tasks ONLY IF practical implementation issues in inference system design are ignored. We selected a specific binary-coding-based quantization to be performed for language models because some practical libraries (e.g., BiQGEMM (Jeon et al., 2020)) exist. Up to date, most (if not all) language models rely on LSTM layers or Transformer-like structures that involve matrix multiplications as the main computation engine. On the other hand, image tasks still depend on various types of convolution layers with a huge amount of features that may be even larger than the size of weights. Then, uniform quantization (rather than non-uniform quantization that we consider in the manuscript) associated with fixed-point representations would be useful for deploying image task models. We have not discussed different characteristics of numerous quantization methods a lot in our manuscript because such discussions have been done in other literature. We believe that Section 2 explains our motivation to consider binary-coding-based quantization and language models specifically for our experiments and detailed analysis.
>
>
> R2: There is a confusion on experimental setting: The quantization should be applied to the fine-tuned model.
>
> A2: We appreciate your careful review. We have conducted all quantization experiments using fine-tuned models only throughout the entire manuscript. In the original manuscript, we intended to explain that all procedures (including fine-tuning and quantization) start from a pre-trained model. To avoid any confusion, we revised section 4.3 to clearly indicate that quantization is performed on the fine-tuned models.
>
>
> R3: It requires exquisite tuning on hyperparameters, which are sensitive as shown in Table 2.
>
> A3: Table 2 shows the motivation for exploring various hyper-parameter sets. In section 5, we show that such an exploration is not challenging because 1) for a grid search, we utilize a pre-arranged fixed 16 combinations of hyper-parameters only, and then 2) BO performs hyper-parameter optimization process automatically. Table 3 confirms that various language models can be quantized by such a simple hyper-parameter search strategy. We revised section 5 to explain the hyper-parameter search process in more detail. In summary, our proposed hyper-parameter tuning can be performed in a structured and efficient manner.
>
>
> R4: Parameters in batch normalization layers?
>
> A4: Our quantization scheme does not quantize activations since quantizing weights only can enhance performance significantly for language models (as discussed in BiQGEMM (Jeon et al., 2020)). Thus, considerations on all kinds of normalization layers are basically optional in our scheme. Moreover, batch normalization is not utilized in the list of language models we have chosen. In Appendix, for image tasks, we did not perform quantization on batch normalization layers that have relatively a lot smaller number of parameters.
>
> Please let us know if you have any other questions or comments.
> Thanks a lot.

---

> > ### Comment · AnonReviewer3 · 2020-11-16
> > **Some further questions**
> >
> > Thanks for your patient response.
> >
> > For your response to R1: I can take that the non-uniform (binary-coding-based) quantization works better for matrix multiplication as shown by BiQGEMM. It is also reluctant to connect the proposed method to language model, since we are in a research field instead of engineering. Besides, convolution on image tasks can also be conducted via matrix multiplication. Anyhow, I take the first question answered.
> >
> > For your response to R3: What I try to address is that the proposed method pay much attention to hyper-parameter search, which should not be a core part for a quantization paper. Besides, the searching is based on BO, which is not a contribution for this paper. The current submission is more like: a published quantization framework + how to determine the detailed parameter + hyper-parameter search.
> >
> > For your response to R4: If no batch normalization is used in BERT? How to deal with the bn layer in original pretrained model? Just ignore them?
> >
> > Besides, I have a key question to the proposed method: through my repeated reading, the quantized value used is represented as $\sum_{i=1}^q \alpha_i b_i$. However in Eq.(6): $M \in \mathbb{R}^{n \times n}$, while $B_q \in \mathbb{R}^q$, that leads to dimension mismatch. I think the core part of the paper is to determine the value in $M$ in Eq.6, which corresponds to $\alpha_i$ earily. But according to " The goal of this work is then to empirically show that magnitude-based importance is indeed practical for post-training binary-coding-based quantization", magnitude-based importance is for each parameters, how is it related to $\alpha_i$ which $\boldsymbol{\alpha} \in \mathbb{R}^q$?
> >
> > There are too many unrelated formula in the paper, such as Eq.7&8. Author should involve these formula in related work of specify them in preliminary. Since your core method is merely to use magnitude-based importance. It leads to a confusing reviewing experience.

---

> > > ### Author Response · Authors · 2020-11-17
> > > **Our response to your questions**
> > >
> > > Thank you so much for your additional comments. We believe our following response can clearly address your concerns.
> > >
> > > **Q1- The proposed method pay much attention to hyper-parameter search, which should not be a core part for a quantization paper.**
> > >
> > > Please understand that we provide a thorough hyper-parameter search process in the paper to enhance *'reproducibility'* of our proposed work. We also wanted to show that a common and simple search process (in Table 5) works for various language models. Our main contribution is to introduce importance metrics for post-training quantization. To our best knowledge, up to date, post-training quantization could not utilize conventional importance metrics based on partial derivatives for post-training quantization. Hence, we suggested and verified that magnitude-based importance (with associated three hyper-parameters) can improve the model accuracy after post-training quantization for the first time.
> > >
> > >
> > > **Q2- No batch normalization is used in BERT? How to deal with the bn layer in original pretrained model?**
> > > We are sorry but this comment is quite confusing for us. Transformers and its variants (including BERT) include layer-norm layers instead of batch-norm layers that are popular for CNN-based models. All models in the experiments do not include batch-norm layers. We believe the implementations and underlying principles of those two normalization methods are quite different. We do not quantize the parameters of layer-norm layers because of a relatively too small number of parameters. Then, because activations are not quantized in our proposed quantization scheme, there are no particular concerns of quantization on layer-norm layers.
> > >
> > >
> > > **Q3- Eq.(6) leads to dimension mismatch**
> > >
> > > We are afraid that there is a misunderstanding in your comment. Since $\mathbf{b} (\in$ {-1,+1}$^n$) is a binary vector as we indicated in the fourth line of Section 3, $\mathbf{B}_q$ follows {-1,+1}$^{n \times q}$. Eq.(6) then produces a vector $\mathbf{\alpha} \in \mathbb{R}^q$. Since the notations of $\mathbf{b}$ and $\mathbf{B}$ might be confusing, we revised Eq.(4) and Eq.(6) in the revised manuscript to show the dimension of $\mathbf{B}_q$ clearly. Please let us know if you still find confusing parts.
> > >
> > >
> > > **Q4- There are too many unrelated formula in the paper**
> > >
> > > For Eq.(7) and (8), please refer to the revised manuscript uploaded. Can you please give us some example equations that are unrelated in your view?

---

> > > > ### Comment · AnonReviewer3 · 2020-11-17
> > > > **Response to response**
> > > >
> > > > Thanks for your clarification.
> > > >
> > > > Q1: I understand your effort to involve magnitude-based importance in post-training quantization, which requires hyper-parameter search in such involvement. But that's not my concern. I am thinking that these searching contributes less to the quantization method in a **research aspect**. For this is research paper instead of technical report. Beside this work is not about BO. Anyhow, I can get your meaning.
> > > >
> > > > Q2: Sorry I make a mistake, what I am refering at the begining should be **layer normalization** instead of batch norm. Now I see that the parameters in layer norm is not quantized.
> > > >
> > > > Q3: The revised version solves my question.
> > > >
> > > > Q4: What I am trying to address is that: Eq.7&8 is from other papers and they are not used in your method. In this case and in my understanding, these 2 formula should appear in related work or preliminary, instead of in the main context, especially where you are going to present your method (Eq.9). Otherwise, it will lead reviewer to consider whether Eq.7&8 are used in the proposed method or not. I understand that you putting here is to emphasize that he gradient or the Hessian are not available in post-training quantization, that's why the proposed method use magnitude-based method. However, from my aspect, it is **unrelated** for listing the formula that are not used in main context.
> > > >
> > > > I think all my questions are answered and solved in certain aspects. And I understand your idea, emphasis and details. Thanks again for your patient response.

---

> > > > > ### Author Response · Authors · 2020-11-19
> > > > > **Many thanks for your quick response!**
> > > > >
> > > > > First, thanks again for your kind response. We are very happy to hear that all of your questions have been answered.
> > > > > We fully respect your comments but we respectively disagree with your comment that our work is somewhat a technical report.
> > > > >
> > > > > **Our search process is NOT cherry-picking. Instead, search results consistently support that weight magnitude is indeed important.**
> > > > >
> > > > > In Table 2 (and Table 6), every single result using our proposed methods (with $E\neq 0$) produces better accuracy than the quantization results of Greedy and Alternating methods (where $E=0$, $C=1$, and $P=0$). On the other hand, if E=0, then fine-tuning of C and P cannot improve model accuracy noticeably compared to the cases of $E\neq 0$. The search results prove that 1) indeed, considering magnitude for post-training quantization is critical to improving model accuracy and 2) once such consideration is accepted, then there is a large search space associated with improved accuracy that is visible only with $E\neq 0$. If our results show high variations on model accuracy (i.e, up and down compared to the plain Greedy or Alternating), then you are absolutely right (it would be just a report of cherry-picking). But even with some variations on hyper-parameters for $E$, $C$, and $P$, model accuracy is **consistently** improved when $E\neq 0$. We believe that our work can enable new post-training quantization research to study practical importance metrics for post-training quantization to enhance model accuracy.
> > > > >
> > > > > We strongly hope that you re-evaluate the message of Table 2 (and Table 6). Our fine-grained search and BO work clearly support our claim that magnitude is a dominating component as an importance metric even when $C$ and $P$ are also useful.

---

> > > > > > ### Comment · AnonReviewer3 · 2020-11-19
> > > > > > **Response to Table 2&6**
> > > > > >
> > > > > > Table 2&6 indeed inform us that with $E=1$, changing $C,P$ affects little to the performance (which means $C,P$ are unsensitive, but as Figure 2 shows, $C, P$ are somehow sensitive). It also show that the importance of magnitude for post-training quantization as indicated by the author.
> > > > > >
> > > > > > Then my next question is: Since author found that $C,P$ contributes less to the performance, why does the author spend much space to study $C, P$ in the main context ? If it is claimed by the author that: "importance metric ($E$) is dominating while $C, P$ are also helpful", then hyper-parameter searching is necessary to find $C, P$.  If $C, P$ is less important, then it comes back to my previous question.
> > > > > >
> > > > > > Overall, my thinking is that: it is unnecessary to use BO to search for $C, P$ (or even not to mention them) if $C, P$ is unsensitive. For BO is not your contribution. If they are really contribute something, Eq.9 (and corresponding BO search) looks more like an **engineering feature** to determine the importance of parameters in post-training quantization.

---

> > > > > > > ### Author Response · Authors · 2020-11-19
> > > > > > > **What would be the main issue with using BO?**
> > > > > > >
> > > > > > >
> > > > > > > If we understand your comments correctly, it seems that you consider our search process (including BO) as a simple engineering effort without new research contributions. We sincerely want to understand why the fact that we used BO makes our work just an engineering attempt (please do not get us wrong, all review comments are very important feedback for us). Let us clarify our (research) contributions related to the search process.
> > > > > > >
> > > > > > > **1. Our work is the first to show that there is a large search space of (binary-coding-based) quantized weights that can be explored by using $E$, $C$, and $P$.**
> > > > > > >
> > > > > > > Previously, the quantized weights for post-training have been obtained by some deterministic methods (such as the Greedy and the Alternating that minimize the quantization error on weights) without further explorations. On the other hand, based on the observations that the minimum quantization error may not lead to the best model accuracy, we break such grounded assumptions and provide new search space that is free from minimum quantization error. Empirical report of new search space (that can be explored in post-training quantization) is one of our MAJOR contributions (that could be explored not only by BO that we use but also by many potential methods in the future research).
> > > > > > >
> > > > > > > **2. As you indicated, BO itself cannot be our contribution.**
> > > > > > >
> > > > > > > Bayesian optimization has been utilized for various machine learning studies for a long time. We cannot claim that BO is our contribution. Optimizing hyper-parameters can be performed by various methods including gradient descent, Newton method, and so on. BO is just one of the useful tools to expedite to optimize parameter search.
> > > > > > >
> > > > > > > **3. We empirically verified that our parameter search can be fine-tuned by BO.**
> > > > > > >
> > > > > > > To enable a BO process, there should be a relationship between a set of {$E$,$C$,$P$} and training accuracy that can be modeled by BO (i.e., if there is a random relationship between a set of hyper-parameters and training accuracy, then BO would not work). During a BO process, we monitor only training accuracy (as discussed in the manuscript), then we check the test accuracy using the set of hyper-parameters associated with the best training accuracy. As such, our experimental results verify that 1) a selection of $E$,$C$, and $P$ and corresponding training accuracy have a relationship that can be modeled by BO and 2) our search process preserves the generalization such that test accuracy is high even if the entire search process is performed by only training accuracy.
> > > > > > >
> > > > > > > We believe that suggesting key hyper-parameters is one of the important research topics in deep learning community. We cannot say that some hyper-parameters are just of engineering efforts when hyper-parameter selection is sensitive to model accuracy. For example, the selections of weight decay factors, dropout rates, and learning rates are very sensitive to model accuracy but it would be difficult to mention that those parameters are of just engineering efforts. We firmly feel that the importance of the introduction of new key hyper-parameters and their sensitivity need to be discussed separately.

---

> > > > > > > > ### Comment · AnonReviewer3 · 2020-11-24
> > > > > > > > **There is nothing wrong with BO**
> > > > > > > >
> > > > > > > > 1. To: "it seems that you consider our search process (including BO) as a simple engineering effort without new research contributions."
> > > > > > > >
> > > > > > > > What I think is (and I repeat again):  Searching contributes less to the **quantization** method in **research aspect**. And quantization should be the core of the paper. $E, C, P$ is the hyper-parameters set in this paper, and their correlation (formulation) should be the **reseach aspect**. Whatever searching method you use to determine their value is the **practical way** to implement the method.
> > > > > > > >
> > > > > > > > 2. To: "why the fact that we used BO makes our work just an engineering attempt"
> > > > > > > >
> > > > > > > > That is because BO is **decoupled** with your proposed method: You may use other searching methods (as you also indicated), and BO can be applied in other methods for hyper-parameters finetuning.
> > > > > > > >
> > > > > > > > 3. To: "hyper-parameters is one of the important research topics in deep learning community":
> > > > > > > >
> > > > > > > > That's for sure. And my point is always that: the searching of hyper-parameters doesn't guide to the design of algorithm, which leads to searching contributes less to the quantization method in a **research aspect**. It is a very important practical process to determine hyper-parameters, especially in deep learning. But that does not conflict my point.
> > > > > > > >
> > > > > > > > Overall, I acknowledge the three keypoints you highlighted in the previous response. And my concern is not about BO or what hyper-parameter searching method you use. My concern is: is $C, P$ sensitive in Eq.9 ? If not, maybe you can focus more on the formulation of post-training quantization (i.e. does other variants of formulation work ?) If yes, then you may use hyper-parameters searching method (BO) for analysis, while the searching part does not contribute to the novelty of the method (but complete the method).

---

### Official Review · AnonReviewer2 · 2020-10-27
**Good work, extends and improves recent approach with novel features**

**Rating:** 6
**Confidence:** 3

**Review:**

The paper employs the binary-coding-based post-training quantization (without retraining) for language modeling. The key contribution is that weight importance is considered while determining binary code (a, B). Two methods, Greedy and Alternating, are also modified to use the importance. The algorithm uses a novel normalized importance, which directly uses weight magnitude and some hyper-parameters. Because the performance is sensitive to hyperparameters, Bayesian optimization is used to find task- and model-specific settings.

Adopting weight importance to previous algorithms is quite novel and makes sense. Binary-coding-based quantization is recently introduced but seems to be quite a promising direction, so this paper has some significance. The idea to replace pruned parameters with the smallest ones is also smart.

The choice of E, C, P as controllable parameters seems reasonable. However, it looks like these parameters are sensitive and differs much task-by-task and model-by-model, that the robustness to the hyper-parameter is not ensured.

Major questions:

1)	I am not sure that in Section 4.1, Equation (7) and (8) properly support the state “weight magnitude can be a dominating factor for the loss function perturbation”. ‘Delta-L’ is a function of both ‘h’ and ‘delta-w’, not ‘w’ itself directly. (I might be misunderstood…)

2)	Can you provide some explanation (or guess)? (a) why sometimes greedy method is better than alter method (b) when does the pruning helps.

Minor comments:

1)	It would be great if a detailed scaling factor sharing scheme inside the parameter is given. For example, row-wise/column-wise/block tiling for a weight matrix?

2)	When the sample dataset (all or some) is not given (that’s why post-training quantization is valuable), how can we select proper hyper-parameters? BO maybe not applicable.

3)	The term “Original” in tables may be misleading that the values are from the floating-point baseline. For example, for Table 1, I recommend clarifying three cases: float baseline, greedy without importance (previously “Original”), greedy with importance (E=1.0).

---

> ### Author Response · Authors · 2020-11-13
> **Response to AnonReviewer2**
>
> We appreciate your positive feedback and careful review.
>
>
> R1: I am not sure Eq. (7) and (8) properly support the state "weight magnitude can be a dominating factor for the loss function perturbation".
>
> A1: We admit that Section 4.1 was quite confusing. To clarify our major contributions and state our assumptions clearly, we revised Section 4.1 to show that a study magnitude-based importance estimation (for post-training quantization) is necessary to replace the Hessian that has been wide-utilized for various sensitivity studies (because the Hessian would not be available for post-training quantization). We acknowledge that it is challenging to connect the magnitude of weights to the loss perturbation in theory. Hence, our contribution should be to 'empirically' verify that magnitude can also be a useful importance estimation when partial derivatives are not available. Accordingly, we conducted various importance-aware quantization methods with numerous language models in order to support our assumptions. It would be an exciting research topic to investigate theories of how such a simple function using magnitude can provide importance metrics for post-training quantization.
>
>
> R2: Can you provide some explanation (a) why sometimes greedy method is better than alter. method (b) when does the pruning helps.
>
> A2(a): Since quantization is basically a non-convex optimization process, depending on the loss landscape, the amount of quantization error may not lead to a minimized training accuracy drop. Hence, even though Greedy method shows a larger weight quantization error compared to Alternating method, training accuracy drop by Greedy method can be less than that of Alternating method.
>
> A2(b): The impact of pruning may not be significant if importance metrics (determined by E) for small weights are already small. We expect the impact of pruning would be noticeable when 1) importance metrics for small weights are relatively high by setting small E and 2) pruning rate is high such that perturbation of small weights does not affect training loss noticeably.
>
>
> R3: A detailed scaling factor sharing scheme inside the parameters?
>
> A3: For all experiments in the manuscript, we selected row-wise quantization that BiQGEMM library (Jeon et al., 2020) also follows. Even though in Section 4.3 we described that \alpha and \mB are computed for each row, we added a sentence to clearly state that row-wise quantization is considered.
>
>
> R4: When the sample dataset (all or some) is not given, how can we select proper hyper-parameters?
>
> A4: Data-free or data-agnostic post-training quantization is also being introduced as a popular post-training quantization method. Data-free quantization, however, does not consider input domains or particular distributions. Even though BO requires sample dataset, as Figure 1 shows, we expect that increasing scaling factors generally improves model accuracy regardless of input domains. It would be an interesting research topic to extend observations of Figure 1 to propose a magnitude-based data-agnostic post-training quantization.
>
>
> R5: The term "Original" in tables may be misleading.
>
> A5: We totally agree with you that "Original" can be misleading. Thank you for this constructive suggestion. Throughout the whole revised manuscript, "Original" is replaced with "No Importance".
>
> We would be happy to answer if you have any other questions or comments.
>
> Thank you so much for reading our response.

---

### Official Review · AnonReviewer4 · 2020-10-27
**Importance weighting integrated into weight quantization**

**Rating:** 6
**Confidence:** 4

**Review:**

This paper proposes a weighted quantization framework that could be applied to general neural networks for language models. One core idea is that different parameters might show different sensitivity towards a loss function change. Thus it would make sense to take use of importance metrics to do weighted quantization across parameters.

Clarity:

The paper is clearly written. With a good introduction of related work and a pretty self-inclusive references to experiment setup. I also like how the paper is organized to motivate the inclusion of weight importance into quantization.

Originality:

Originality in this paper is mostly from introducing the concept of parameter (weight) importance and how it is defined and applied to the quantization problem.
This paper is based on an earlier paper (also new), e.g., BiQGEMM (Jeon et al., 2020), which paved the foundation  to support binary-coding-based quantization techniques to accelerate quantized neural networks. A magnitude-based importance metric is proposed and approved effective in the form of the binary codes. To fine-tune model accuracy, three hyper-parameters are explored and empirically investigated to discover best performance (regarding model quality).

Significance:

Based on the experiment result, I agree that this paper's contribution is significant. If we combine the effort of quantization with distillation, as shown with the metrics in DistillBert, we could achieve great ability of inference speed, a very smaller model size but still maintain reasonable model quality.

---

> ### Author Response · Authors · 2020-11-13
> **Response to AnonReviewer4**
>
> We appreciate your positive feedback.
> We are happy to see that our contributions are well recognized in your review.
> We totally agree that combining distillation and quantization (and potentially also structured/unstructured pruning and/or low-rank approximation) would be an important and exciting research direction since the size of language models is substantially increasing.

---

### Official Review · AnonReviewer1 · 2020-10-28
**Inconsistent loss formulation and importance indicator and limited novelty. Lack of comparison with a similar post-training quantization method for NLP tasks.**

**Rating:** 4
**Confidence:** 5

**Review:**

Summary:
- This paper proposes to consider the importance of each parameter in the post-training quantization of weights. The authors propose to use weight magnitude as the importance indicator and to minimize the weighted distance between the full-precision weights and quantized weights. Experiments are performed on various NLP models and tasks.

Strengths:
- The paper is structured clearly and the proposed method is simple.

Weaknesses:
1. It is not clear to me why the change in the loss in (7) and (8) is necessarily related to the magnitude of the weights. From (8), the loss change \delta L  is only related to the hessian, and each weight's perturbation, but not the weight's magnitude.

2.  The second concern comes from the novelty of the proposed method. Indeed, loss-aware (or importance-aware as in this paper) quantization which considers the quantization effect of each parameter to the loss has already been proposed several years ago in [1,2]. In [1,2], they also approximate the loss using the second-order Taylor expansion like in equation (7) in this submission. Moreover, the proposed importance-aware quantization solution in equation (5)  is exactly the same as equation (8) in [1], except that their importance is derived from the diagonal hessian instead of weight magnitude (which is not quite reasonable, refer to the first point).

3. One other post-training quantization method GOBO in [3], which also uses codes and codebooks to quantize language models, is also not compared. From their reported results, under the same number of bits, GOBO has higher accuracy than the proposed method. E.g., GOBO has 83.76% accuracy for the 3-bit quantization on MNLI  while the proposed method only has 82.9%.

4. From Table 2, there is no winning configuration of (E,C,P) that works well on all studied models and tasks. How to determine these hyperparameters for  new tasks empirically? Tuning these parameters separately for each task can be inefficient.

Reference:

[1] Hou et al. "Loss-aware binarization of deep networks." International Conference on Learning Representations. 2017.

[2] Hou et al. "Loss-aware weight quantization of deep networks." International Conference on Learning Representations. 2017.

[3] Zadeh, Ali Hadi, and Andreas Moshovos. "GOBO: Quantizing Attention-Based NLP Models for Low Latency and Energy Efficient Inference." arXiv:2005.03842, 2020.

---

> ### Author Response · Authors · 2020-11-13
> **Response to AnonReviewer1 (Part 1/2)**
>
> We highly appreciate your time and efforts to review our manuscript.
>
> R1: It is not clear why the change in the loss in (7) and (8) is necessarily related to the magnitude of the weights.
>
> A1: We admit that Section 4.1 was quite confusing. To clarify our major contributions and state our assumptions clearly, we revised Section 4.1 to show that a study magnitude-based importance estimation (for post-training quantization) is necessary to replace the Hessian that has been wide-utilized for various sensitivity studies (because the Hessian would not be available for post-training quantization). We acknowledge that it is challenging to connect the magnitude of weights to the loss perturbation in theory. Hence, our contribution should be to 'empirically' verify that magnitude can also be a useful importance estimation when partial derivatives are not available. Accordingly, we conducted various importance-aware quantization methods with numerous language models in order to support our assumptions. It would be an exciting research topic to investigate theories of how such a simple function using magnitude can provide importance metrics for post-training quantization.
>
>
> R2: Loss-aware (or importance-aware as in this paper) quantization which considers the quantization effect of each parameter to the loss has already been proposed several years ago in [1,2].
>
> A2: We acknowledge that loss-aware quantization has been widely utilized for the compression-aware training method. On the other hand, post-training quantization has serious challenges to produce sensitivity metrics because partial derivatives would not be available once training is completed (inference codes do not include backward propagation descriptions or optimization steps using gradient descent). We believe that is why there are few studies to find importance metrics for post-training quantization in the past because most importance or sensitivity estimations have been deeply related to the 1st or 2nd partial derivatives (e.g, the gradients or the Hessian). Thus, in the manuscript, we put a bold assumption that a function using magnitude only can be also used as an importance estimation in the case of post-training quantization. Of course, such an assumption needs to be empirically supported as we demonstrated throughout various experiments in the manuscript. We revised Section 4.1 accordingly.
>
> Eq. (5) and (6) in the manuscript are fundamentally different from the solutions introduced in [1] and [2]. For example, reference [1] considers binary neural networks where an analytical solution for quantization exists while we propose various importance-aware approximation methods in Eq. (5) and (6) for multi-bit quantization. Reference [2] assumes ternary or symmetric uniform quantization where the number of scaling factors would be the same as that of binary neural networks. Overall, reference [1] and [2] are vastly different from ours because 1) [1] and [2] are not post-training methods, 2) [1] and [2] study compression-aware training methods where partial derivatives (e.g., the Hessian) can be derived, 3) quantization formats are different from ours such that implementation principles are also different from ours.

---

> > ### Author Response · Authors · 2020-11-13
> > **Response to AnonReviewer1 (Part 2/2)**
> >
> >
> > R3: One other post-training quantization method GOBO in [3] is also not compared.
> >
> > A3: Thank you for introducing an interesting paper. We admit that there are a lot of exciting quantization schemes with different implementation considerations. We believe that comparing our results with quantization method of other forms (such as GOBO) would not be fair because 1) GOBO is proposed as a cross-disciplinary approach in order to design a new chip architecture to support codebook-based quantization while binary-coding-based quantization in our manuscript can be operated by commercialized general-purpose chips (like CPUs and GPUs as discussed in BiQGEMM (Jeon et al., 2020)), and 2) in terms of model accuracy, codebook-based quantization is inherently superior to binary-coding-based quantization that has a restricted form to facilitate computational efficiency (basically the impact of codebook-based quantization is only memory storage/bandwidth reduction while binary-coding-based quantization simplifies computations without complicated dequantization process during inference). Because the goal of this study is not only to show improved quantization algorithms but also to present a systematic approach to running inference with known efficient libraries or kernels, we included Section 2 which explains why we limit our attention to language models and a specific quantization form. We believe that if someone wants to compare our proposed technique with quantization methods of different forms (i.e., other than binary-coding-based one), it would be necessary to discuss a lot of related parts including a list of required hardware supports, different benefits at system-level design, and so on.
> >
> >
> > R4: How to determine these hyperparameters for new tasks empirically? Tuning these parameters separately for each task can be inefficient.
> >
> > A4: Our goal was to present a common and systematic way to find hyper-parameters conveniently even for new tasks. Certainly, Table 2 shows the motivation for exploring various hyper-parameter sets. Then, in Section 5, we show that such an exploration is not challenging because 1) for a grid search, we utilize a pre-arranged fixed 16 combinations of hyper-parameters only, and then 2) BO performs hyper-parameter optimization process automatically. Table 3 confirms that various language models can be quantized by such a simple hyper-parameter search strategy. We revised Section 5 to explain the hyper-parameter search process in more detail. In summary, our proposed hyper-parameter tuning is a structured and efficient technique that can be widely applied to various language models.
> >
> >
> > We would be happy to answer any additional questions.
> >
> > Let us know if there are additional ways to raise your score.
> >
> > Thank you so much.

---

> > > ### Comment · AnonReviewer1 · 2020-11-18
> > > **Some further comments**
> > >
> > > Thanks very much for the detailed feedback.
> > >
> > > For your response to R1, indeed in [1,2], the authors use the moving average of the second moment in Adam optimizer to estimate the diagonal of the Hessian. After training, one can still use this moving average as an estimation of the importance of weights. Though the revised version emphasizes that the submission focuses on empirically verifying that using weight magnitude as an importance metric helps post-training quantization, it would make the work more solid to theoretically or mathematically explain why the magnitude of the weights can act as a good importance indicator.
> > >
> > > For your response to R2, as said in the previous paragraph,  the approximate diagonal of Hessian from [1,2] can be saved and used for post-training quantization. In equation (8) in loss-ware binarization [1], ||d_l^{t-1}||_1 = \sum_{i=1}^n d_i, and (8) is exactly the same as the solution for binarization in equation (5) in this submission, except that you use w while they use d to denote the importance of the weights.
> > >
> > > For your response to R3,  GOBO does not require to "design a new chip architecture to support codebook-based quantization". Indeed,  GOBO memory compression mechanism is plug-in compatible with many architectures like TPU, Eyeriss, and architectures using Tensor Cores-like units. Please check the updated paper https://arxiv.org/pdf/2005.03842.pdf for speedup in various hard platforms.
> > >
> > > For your response to R4, still, it does not make much sense for an algorithm to search three different hyperparameters for each task. This also indicates that the proposed model is not very robust and is very sensitive to these parameters. Moreover, what is the computational complexity of the BO process?

---

> > > > ### Author Response · Authors · 2020-11-19
> > > > **We are struggling to catch your point**
> > > >
> > > > First of all, we appreciate your detailed comments and discussions. Certainly, the references you shared are publications of high quality in the areas of compression-aware training and codebook-based quantization. But unfortunately, we failed to fully understand why the ideas on those references are needed to be discussed for the review of our manuscript. Let us show some examples of why we feel so.
> > > >
> > > > **1) [1,2] assume that an optimizer using the 2nd-moment values is given.**
> > > >
> > > > For [1,2], $||d_l^{t-1}||_1 = \sum _{i=1}^n d_i$ is given not only for calculating scaling factors but also for updating weights through the (iterative) proximal Newton method (for training). Correspondingly in [1,2], the entire training procedures (from Adam) are modified to adopt a curvature matrix as an approximated Hessian matrix. While such a method can be useful for compression-aware training, we DO NOT assume that any particular optimizer is already provided (modifying the optimizer is not considered, either). For example, if an SGD optimizer or a similar one with the 1st-moment only is employed for training, then the procedures of [1,2] would not be available (even for compression-aware training). For our work, we do not want to depend on a specific optimizer structure where importance needs to be tightly coupled with a weight update process as shown in [1,2]. We feel that even though [1,2] present good quantization-aware training algorithms, underlying principles of [1,2] would not work for post-training quantization where weight update is not allowed.
> > > >
> > > > **2) As [1,2] show, the Hessian calculation is difficult (if not impossible for post-training quantization) and checkpoints from pre-trained models usually do not involve optimizer-related info.**
> > > >
> > > > As [1,2] clearly indicate, computing the exact Hessian is highly challenging. We acknowledge that there are a few efficient techniques to compute the approximated diagonal Hessian values, but it is difficult (at least for us) to assume that many pre-trained models already include relevant information to conveniently extract the approximated Hessian. For example, we used checkpoints provided by HuggingFace and there are no 2nd-moment values (nor the Hessian) in the checkpoint files. If there are existing publications that successfully leverage the Hessian-like values to enhance post-training quantization (with a successful approximation scheme for the Hessian without special assumptions on the training algorithm), then it would be interesting to compare our work with those ideas. We still believe that using magnitude as importance is better (if not the best) practical technique for post-training quantization than using the Hessian (that would require special care and assumptions for training).
> > > >
> > > > **3) Eq(8) in [1] is fundamentally different from our equations with importance**
> > > >
> > > > In the case of binary neural networks (i.e., 1-bit quantization), it is not difficult to derive Eq(8) because there is an easy analytical solution. But in the case of multi-bit quantization, it is known that there are no analytical solutions. Instead, minimizing quantization error needs to be performed heuristically (like the Greedy method) or iteratively (like the Alternating method). In Section 3.2, we successfully modified both equations (for the Greedy and Alternating) to employ importance metrics (unlike [1,2], those equations are only for one-time post-training quantization, not for weight updates). [1,2] discuss 1-bit quantization or uniform quantization that are not related to Section 3.2 in our manuscript.
> > > >
> > > > **4) GOBO is for codebook-based quantization while we follow the binary-coding-based quantization**
> > > >
> > > > We admit that GOBO has its own merit, especially high accuracy compared to the binary-coding-based quantization. But as we wrote in the manuscript, the binary-coding-based quantization allows significantly simplified computations for MatMul (due to restricted quantization forms). For example, BiQGEMM (dedicated to the binary-coding-based quantization) enables a lot simplified and fast computations when implemented by **CPUs or GPUs** without multiplications. In contrast, codebook-based quantization focuses on improving memory compression capability instead of efficient computational structures. Because those two quantization methods have different assumptions and application areas, we compared our work with only binary-coding-based publications.

---

> > > > > ### Author Response · Authors · 2020-11-19
> > > > > **.. Continued..**
> > > > >
> > > > >
> > > > > **5) The proposed method is not very robust and very sensitive to parameters**
> > > > >
> > > > > In Table 2 (and Table 6), every single result using our proposed methods (with $E\neq 0$) produces better accuracy than the quantization results of Greedy and Alternating methods (where $E=0$, $C=1$, and $P=0$). On the other hand, if $E=0$, then fine-tuning of $C$ and $P$ cannot improve model accuracy noticeably compared to the cases of $E\neq 0$. The search results prove that 1) indeed, considering magnitude for post-training quantization is critical to improving model accuracy and 2) once such consideration is accepted, then there is a large search space associated with improved accuracy that is visible only with $E\neq 0$. If our results show high variations on model accuracy (i.e, up and down compared to the plain Greedy or Alternating), then you are absolutely right (it would be just a report of cherry-picking). But even with some variations on hyper-parameters for $E$, $C$, and $P$, model accuracy is **consistently** improved when $E\neq 0$. We believe that our work can enable new post-training quantization research to study practical importance metrics for post-training quantization to enhance model accuracy.
> > > > >
> > > > > We strongly hope that you re-evaluate the message of Table 2 (and Table 6). Our fine-grained search and BO work clearly support our claim that magnitude is a dominating component as an importance metric even when $C$ and $P$ are also useful.

---

### Author Response · Authors · 2020-11-13
**Revised manuscript is uploaded**

We thank all the reviewers for their valuable comments and detailed feedback. Please find an uploaded revised manuscript. We revised the manuscript according to the reviewers' comments to address concerns and to avoid any unnecessary confusion. All changes in the revised manuscript are highlighted in blue.

---

### Decision · Program_Chairs · 2021-01-07
**Final Decision**

**Decision:**

Reject

**Comment:**

The paper builds upon a recent paper BiQGEMM, providing a binary coding based post training quantization technique. The authors show how to combine magnitude-based importance metrics to these techniques and achieve superior performance. The use of importance metrics for quantization and pruning is not new, and magnitude-based metrics are among the more common metrics. With that in mind, the novelty of the paper is in the integration of importance metrics to the techniques of BiQGEMM. The provided methods lead to several hyper-parameters and the task of tuning these can be non-trivial and time consuming. Due to this the authors devote a detailed section showing how to properly tune these hyper-parameters. This is appreciated and indeed alleviates the problem coming with new hyper-parameters.

The paper received mixed opinions by the reviewers related to its overall novelty, but the resulting conclusion is that although the combination of binary coding based quantization with importance scores is not trivial, the challenges faced relate more to correct implementation as opposed to scientific novelty. Combined with other issues raised by the reviewers such as a need for further comparison with existing work, this lead me to recommend rejection for this paper.